# MOS: Model Synergy for Test-Time Adaptation on LiDAR-Based 3D Object Detection

**Zhuoxiao Chen**[†]    **Junjie Meng**[†]    **Mahsa Baktashmotlagh**[†]    **Yonggang Zhang**[‡]
**Zi Huang**[†]    **Yadan Luo**[†*]
[†]The University of Queensland    [‡]Hong Kong Baptist University

## Abstract

LiDAR-based 3D object detection is crucial for various applications but often experiences performance degradation in real-world deployments due to domain shifts. While most studies focus on cross-dataset shifts, such as changes in environments and object geometries, practical corruptions from sensor variations and weather conditions remain underexplored. In this work, we propose a novel online test-time adaptation framework for 3D detectors that effectively tackles these shifts, including a challenging *cross-corruption* scenario where cross-dataset shifts and corruptions co-occur. By leveraging long-term knowledge from previous test batches, our approach mitigates catastrophic forgetting and adapts effectively to diverse shifts. Specifically, we propose a Model Synergy (MOS) strategy that dynamically selects historical checkpoints with diverse knowledge and assembles them to best accommodate the current test batch. This assembly is directed by our proposed Synergy Weights (SW), which perform a weighted averaging of the selected checkpoints, minimizing redundancy in the composite model. The SWs are computed by evaluating the similarity of predicted bounding boxes on the test data and the independence of features between checkpoint pairs in the model bank. To maintain an efficient and informative model bank, we discard checkpoints with the lowest average SW scores, replacing them with newly updated models. Our method was rigorously tested against existing test-time adaptation strategies across three datasets and eight types of corruptions, demonstrating superior adaptability to dynamic scenes and conditions. Notably, it achieved a 67.3% improvement in a challenging cross-corruption scenario, offering a more comprehensive benchmark for adaptation. Source code: https://github.com/zhuoxiao-chen/MOS.

## 1 Introduction

Recently, LiDAR-based 3D object detectors have exhibited remarkable performance alongside a diverse array of applications such as robotic systems (Ahmed et al., 2018; Montes et al., 2020; Zhou et al., 2022; Ye & Qian, 2018) and self-driving cars (Deng et al., 2021; Wang et al., 2020a; Luo et al., 2023b; Qian et al., 2022; Wang et al., 2019; Chen et al., 2023b; Arnold et al., 2019; You et al., 2020; Meyer et al., 2019; Li et al., 2019; McCrae & Zakhor, 2020; Luo et al., 2023a; Zhang et al., 2024). However, when deployed in the real world, the detection system could fail on unseen test data due to unexpectable varying conditions, which is commonly referred to as the *domain shift*. To study this shift, researchers consider cross-dataset scenarios (*e.g.*, from nuScenes (Caesar et al., 2020) to KITTI (Geiger et al., 2012)) where object sizes and beam numbers are varying in the test domain (Ganin et al., 2016; Luo et al., 2021; Wei et al., 2022), or adding perturbations to clean datasets (*e.g.*, from KITTI to KITTI-C (Kong et al., 2023)) to mimic realistic corruptions caused by severe weather condition and sensor malfunctions (Hahner et al., 2022; Kong et al., 2023; Dong et al., 2023). However, in real-world scenarios, shifts generally do not arise from a unitary source, for example, considering a detection system deployed in an unfamiliar northern city, it is likely to suffer severe weather conditions, such as heavy snow. This leads us to consider a new hybrid shift termed **cross-corruption** (illustrated in Fig. 1), where cross-dataset gaps and corruption noises coexist across 3D scenes (*e.g.*, Waymo (Sun et al., 2020) to KITTI-C).

---

[*]Correspondence to Yadan Luo <y.luo@uq.edu.au>.

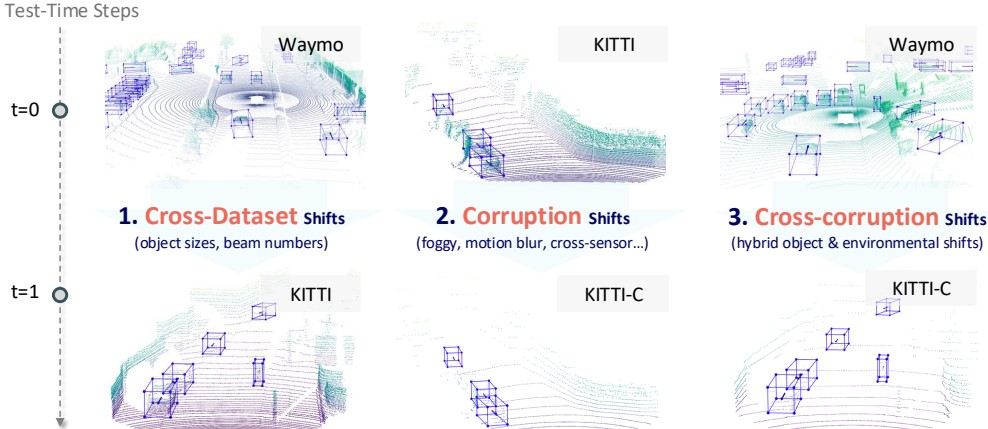

Figure 1: We investigate *three* distinct types of domain shifts that 3D detectors face during test time. Unlike previous work that focuses *solely* on cross-dataset shifts, our study comprehensively explores shifts arising from varying weather conditions, sensor corruptions, and the most challenging cross-corruption shifts, which encompass both object- and environment-related variations.

These simulated shifts prompt a critical research question: *how can the 3D detector naturally adapt to a shifted target scene?* Previous studies have identified unsupervised domain adaptation (UDA) as a promising solution, by multi-round self-training on the target data with the aid of pseudo-labeling techniques (Chen et al., 2023a; Yang et al., 2021; 2022; Peng et al., 2023; Li et al., 2023a), or forcing feature-level alignment between the source and target data for learning domain-invariant representations (Luo et al., 2021; Chen et al., 2021; Zhang et al., 2021; Luo et al., 2023c; Zeng et al., 2021). While effective, these UDA-based approaches require an offline training process over **multiple epochs** with pre-collected target samples. Considering a detection system being deployed on a resource-constrained device in the wild, performing such a time-consuming adaptation during testing is **impractical**. To this end, there is an urgent need for a strategy that allows models to adapt to live data streams and provide instant predictions. Test-time Adaptation (TTA) emerges as a viable solution and has been explored in the general classification task, through (1) choosing a small set of network parameters to update (*e.g.*, matching BN statistics (Wang et al., 2021; Niu et al., 2023)) or (2) employing the mean-teacher model to provide consistent supervision (Wang et al., 2022a; 2023b; Yuan et al., 2023; Tomar et al., 2023a).

A recent work of MemCLR (VS et al., 2023), extends the idea of TTA to image-based 2D object detection by extracting and refining the region of interest (RoI) features for each detected object through a transformer-based memory module.

Despite TTA's advances in image classification and 2D detection, adapting 3D detection models at test time remains **unexplored**. This motivates us first to conduct a pilot study and investigate the effectiveness of these TTA techniques when applied to 3D detection tasks. As plotted in Fig. 2, methods that employ the mean-teacher model (*e.g.*, MemCLR, and CoTTA) outperform the rest

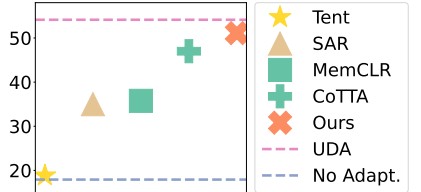

Figure 2: The results ($AP_{3D}$) of applying existing TTA methods to adapt SECOND (Yan et al., 2018) from nuScenes to KITTI. Mean-teacher models are in green.

of the methods. This superiority stems from the teacher model's ability to accumulate long-term knowledge, offering stable supervision signals and reducing catastrophic forgetting across batches (Kirkpatrick et al., 2017; Hayes et al., 2020; Hu et al., 2019; Lee et al., 2017; Kemker et al., 2018). However, by treating all previous checkpoints *equally* through moving average (EMA) for each test sample, the teacher model may fail to recap and leverage the most critical knowledge held in different checkpoints. For example, Fig. 3 shows very different characteristics of test point clouds $x_{t+1}$ and $x_{t+2}$, while the teacher model still relies on a uniform set of knowledge (*i.e.*, generated by EMA), without leveraging relevant insights from prior checkpoints tailored to each specific scene. Hence, no teacher model perfectly fits all target samples. We argue that an optimal "super" model should be dynamically assembled to accommodate each test batch.

In this paper, we present a Model Synergy (MOS) strategy that adeptly selects and assembles the most suitable prior checkpoints into a unified super model, leveraging long-term information to guide the supervision for the current batch of test data. To this end, we introduce the concept of synergy weights (SW) to facilitate the model assembly through a weighted averaging of previously trained checkpoints. A model synergy bank is established to retain $K$ previously adapted checkpoints. As illustrated in Fig. 3, to assemble a super model tailored to test batch $x_{t+1}$, checkpoints 1 and 3 are assigned lower SW due to redundant box predictions. To leverage the knowledge uniquely held by checkpoint 4 (*e.g.*, concepts about cyclists and pedestrians), MOS assigns it with a higher SW.

To determine the optimal SW for the ensemble, we reformulate this as solving a linear equation, intending to create a super model that is both diverse and exhibits an equitable level of similarity to all checkpoints, without showing bias towards any. The closed-form solution of this linear equation leads to calculating the inverse Gram matrix, which quantifies the similarity between checkpoint pairs. To this end, we devise two similarity functions tailored for 3D detection models from 1) *output-level*, measuring prediction discrepancies through Hungarian matching cost of 3D boxes, and 2) *feature-level*, assessing feature independence via calculating the matrix rank of concatenated feature maps. By computing these similarities for each new test batch, we determine the SW for the construction of the super model. This super model then generates pseudo labels, which are employed to train the model at the current timestamp for a single iteration.

However, with more saved checkpoints during test-time adaptation, the memory cost significantly increases. To address this, we introduce a dynamic model update strategy that adds new checkpoints, while simultaneously removing the least important ones. Empirical results evidence that our approach, which maintains only 3 checkpoints in the model bank, outperforms the assembly of the 20 latest checkpoints and reduces memory consumption by 85%. The **contributions** of this paper are:

1. This is an early attempt to explore the test-time adaptation for LiDAR-based 3D object detection (TTA-3OD). In addressing the challenges inherent to TTA-3OD, we propose a novel Model Synergy (MOS) approach to dynamically leverage and integrate collective knowledge from historical checkpoints.

2. Unlike mean-teacher based methods that aggregate all previous checkpoints, we identify and assemble the most suitable ones according to each test batch. To this end, we utilize the inverse of the generalized Gram matrix to determine the weights. We introduce similarity measurements for 3D detection models at both feature and output levels to calculate the generalized Gram matrix.

3. We conduct comprehensive experiments to address real-world shifts from (1) cross-dataset, (2) corrupted datasets, and (3) hybrid cross-corruption scenarios including eight types of simulated corruptions. Extensive experiments show that the proposed MOS even outperforms the UDA method, which requires multi-round training, when adapting from Waymo to KITTI. In a more challenging scenario of cross-corruption, MOS surpasses the baseline and direct inference by 67.3% and 161.5%, respectively.

## 2 RELATED WORK

**Domain Adaption for 3D Object Detection** aims to transfer knowledge of 3D detectors, from labeled source point clouds to an unlabeled target domain, by mitigating the domain shift across different 3D scenes. The shift in 3D detection is attributed to variations in many factors such as object statistics (Wang et al., 2020b; Tsai et al., 2023b), weather conditions (Xu et al., 2021; Hahner et al., 2022), sensor differences (Rist et al., 2019; Gu et al., 2021; Wei et al., 2022), sensor failures (Kong et al., 2023), and the synthesis-to-real gap (Saleh et al., 2019; DeBortoli et al., 2021; Lehner et al., 2022). To bridge these gaps, approaches including adversarial matching of features (Zhang et al., 2021), the generation of enhanced 3D pseudo labels (Yang et al., 2021; 2022; Chen et al., 2023a; Saltori et al., 2020; Wang et al., 2022b; You et al., 2022; Li et al., 2023b; Wang et al., 2023c; Peng et al., 2023; Huang et al., 2024; Tsai et al., 2023a), the mean-teacher model (Luo et al., 2021; Hegde et al., 2023) for stable adaptation, and contrastive learning (Zeng et al., 2021; Lim et al., 2024) for tighter embeddings have been extensively investigated. However, implementing these cross-domain methods necessitates multiple epochs of retraining, rendering them impractical for scenarios where test data arrives in a streaming manner.

**Test-Time Adaptation (TTA)** seeks to adapt the model to unseen data at inference time, by addressing domain shifts between training and test data (Wang et al., 2024b; Chen et al., 2024). A very first

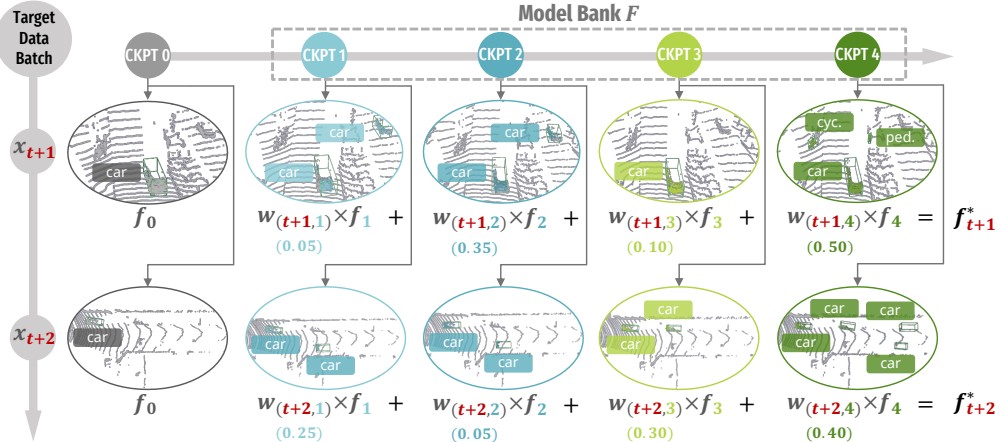

Figure 3: Illustration of the **model synergy (MOS)** that selects key checkpoints and assembles them into **a super model** $f_{t+1}^*$ **that tailors for each of test data** $x_{t+1}$. "cyc." denotes the cyclist and "ped." denotes the pedestrian. MOS prioritizes checkpoints with *unique* insights that are absent in other checkpoints with *higher* weights while reducing the weights of those with redundant knowledge.

work, Tent (Wang et al., 2021) leverages entropy minimization to adjust batch norm parameters. Subsequently, many works have adopted similar approaches of updating a small set of parameters, each with different strategies and focus (Niu et al., 2022; Mirza et al., 2022; Gong et al., 2022; Yuan et al., 2023; Zhao et al., 2023; Niloy et al., 2024; Gong et al., 2024; Wang et al., 2024a; Niu et al., 2023; Hong et al., 2023; Song et al., 2023; Nguyen et al., 2023; Niloy et al., 2024). For instance, EATA (Niu et al., 2022) identifies reliable, non-redundant samples to optimize while DUA (Mirza et al., 2022) introduces adaptive momentum in a new normalization layer. NOTE (Gong et al., 2022) updates batch norms at the instance level, whereas RoTTA (Yuan et al., 2023) and DELTA (Zhao et al., 2023) leverage global statistics for batch norm updates. Additionally, SoTTA (Gong et al., 2024) and SAR (Niu et al., 2023) enhance batch norm optimization through sharpness-aware minimization. Alternatively, some approaches optimize the entire network through the mean-teacher framework for stable supervision (Wang et al., 2022a; Tomar et al., 2023b), generating reliable pseudo labels for self-training (Goyal et al., 2022; Zeng et al., 2024), employing feature clustering (Chen et al., 2022; Jung et al., 2023; Wang et al., 2023b), and utilizing augmentations to enhance model robustness (Zhang et al., 2022). Despite these TTA methods being developed for general image classification, MemCLR (VS et al., 2023) is the pioneering work applying TTA for image-based 2D object detection, utilizing a mean-teacher approach for aligning instance-level features. Nevertheless, the applicability of these image-based TTA methods to object detection from 3D point clouds remains unexplored.

## 3 PROPOSED APPROACH

**Overall Framework.** First, we establish the definition and notations for the Test-Time Adaptation for 3D Object Detection (TTA-3OD). When deploying a 3D detection model pretrained on a source dataset, the test point clouds $\{x_t\}_{t=1}^T$ are shifted or/and corrupted due to varying real-world conditions, where $x_t$ is the $t$-th test point cloud in the stream of test data. The ultimate goal of TTA-3OD is to adapt the 3D detection model to a sequence of target scenes in a single pass during the inference time.

The core idea of our methodology is to identify the most suitable historical models with distinct knowledge **for each test batch** $x_t$, and assemble them into a **super model** $f_t^*$ facilitating the model adaptation during the test time. To this end, we establish a model bank $\mathbf{F} = \{f_1, \ldots, f_K\}$ of size $K$ to preserve and synergize the previously trained models. The overall workflow of our method unfolds in three phases, as shown in Fig. 4 and Algorithm 1 (in Appendix).

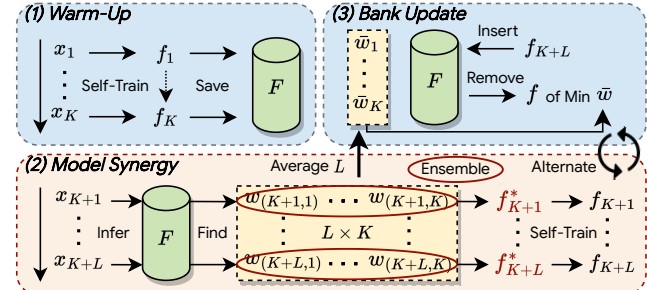

Figure 4: The overall workflow of our approach.

**Phase 1: Warm-up** Initially, when the model bank $\mathbf{F}$ is not yet populated, we train the model with the first $K$ arriving target batches by pseudo labeling-based self-training. We save the model as a checkpoint at each timestamp, $\{f_1, \cdots, f_K\}$, into the $\mathbf{F}$ until reaching its capacity of $K$.

**Phase 2: Model Synergy** Once $\mathbf{F}$ is full, for every subsequent test batch $x_t$, checkpoints within $\mathbf{F}$ are assembled into a super model $f_t^*$ to provide pseudo labels $\hat{\mathbf{B}}^t$ for supervision. For instance, as illustrated in Phase 2 of Fig. 4, when inferring $x_{K+1}$, we assemble the super model via $\mathbf{F}$ into $f_{K+1}^* = w_{(K+1,1)}f_1 + \cdots + w_{(K+1,K)}f_K$. Then, pseudo labels $\hat{\mathbf{B}}^{K+1}$ are generated by $f_{K+1}^*$ to train the current model $f_{K+1}$ for a single iteration. When $L$ batches are tested, we update the model bank.

**Phase 3: Model Bank Update** To maintain a compact model bank $\mathbf{F}$, meanwhile, each model holds distinct knowledge, we dynamically update the bank $\mathbf{F}$ to fix its size at $K$. For every $L$ batches tested, we replace the checkpoint with the most redundant knowledge in $\mathbf{F}$, with the newly trained $f_t$. Phases 2 and 3 alternate until all target samples $\{x_t\}_{t=1}^T$ have been tested.

### 3.1 Model Synergy (Phase 2)

As Phase 1 is straightforward, where the model bank is filled up with the first $K$ checkpoints, we elaborate on the details of Phase 2 in this subsection. An intuitive example is presented in Fig. 3: when inferring a target point cloud, different historical models yield inconsistent predictions because these models learned characteristics from different batches at earlier timestamps. However, if these characteristics no longer appear in subsequent arriving test data, the model may suffer catastrophic forgetting, losing previously learned information. Our strategy to overcome this is to leverage and combine the long-term knowledge from prior checkpoints. But, not all checkpoints are equally important, as shown in Fig. 3, checkpoints 1 and 2 produce highly similar box predictions, whereas checkpoint 4 can detect objects that are missed by others. Treating these checkpoints equally leads to the accumulation of redundant information. Therefore, we propose synergy weights (SW) to emphasize unique insights and minimize knowledge redundancy, via weighted averaging of the historical checkpoints. Ideally, similar checkpoints (*e.g.*, CKPT 1 and 2 in Fig. 3) should be assigned with *lower* SW, whereas the checkpoint exhibits unique insights (*e.g.*, CKPT 4 in Fig. 3) should be rewarded with *higher* SW. Through weighting each checkpoint, we linearly assemble them into a super model $f_t^*$ that best fits the current test batch $x_t$. In the following section, we discuss our approach to determine these optimal SW for the super model ensemble.

#### 3.1.1 Super Model Ensemble by Optimal Synergy Weights

For each batch $x_t$, we assume there exist optimal synergy weights $\mathbf{w} \in \mathbb{R}^K$ for linearly assembling the historical checkpoints within the model bank $\mathbf{F}$ into a super model $f_t^*$ as below:

$$\mathbf{F}\mathbf{w} = f_t^*. \tag{1}$$

For simplicity, we use the same notation $f_t$ to indicate the model parameters. The optimal synergy weights $\mathbf{w}$ can be deduced by solving the linear equation:

$$\mathbf{w} = (\mathbf{F}^T\mathbf{F})^{-1}\mathbf{F}^T f_t^*, \tag{2}$$

which can be decomposed into two parts, (1) $(\mathbf{F}^T\mathbf{F})^{-1}$: inverse of the Gram matrix, and (2) $\mathbf{F}^T f_t^*$: pairwise similarities between each model in $\mathbf{F}$ and the super model $f_t^*$. Ideally, these similarities are expected to be uniformly distributed, ensuring $f_t^*$ remains unbiased towards any single model in $\mathbf{F}$ for a natural fusion of diverse, long-term knowledge. Thus, we can simplify the Eq. (2) as:

$$\mathbf{w} = (\mathbf{F}^T\mathbf{F})^{-1}\mathbb{1}^K. \tag{3}$$

Now, to compute the optimal synergy weights $\mathbf{w}$, we delve into the first part of Eq. (3) to compute the Gram matrix $\mathbf{G} = \mathbf{F}^T\mathbf{F}$ which captures the model parameter similarity for any arbitrary pair of models $(f_i, f_j)$ in the bank $\mathbf{F}$, as:

$$\mathbf{G} = \mathbf{F}^T\mathbf{F} = \left(\langle f_i, f_j \rangle\right)_{i,j=1}^K \in \mathbb{R}^{K \times K}, \ f_i, f_j \in \mathbf{F}, \tag{4}$$

where $\langle f_i, f_j \rangle$ denotes the inner product of the model parameters of $f_i$ and $f_j$. Based on the similarities captured in $\mathbf{G}$, its inverse, $\mathbf{G}^{-1}$, essentially prioritizes directions of higher variance (diverse information) and penalizes directions of lower variance (duplicated information). Thus, the synergy weights calculated by $\mathbf{w} = \mathbf{G}^{-1}\mathbb{1}^K$, are theoretically low for those checkpoints with knowledge duplicates. Previous studies (Dinu et al., 2023; Yuen, 2024; Nejjar et al., 2023) have also demonstrated the effectiveness of using the $\mathbf{G}^{-1}$ to determine importance weights.

### 3.1.2 INVERSE OF GENERALIZED GRAM MATRIX FOR 3D DETECTION MODEL

While the inverse of the Gram matrix is effective, evaluating similarity through the inner product of model parameters $f_i$ and $f_j$ is suboptimal, as models with *distinct* parameters may produce *similar* predictions on the same input test batch $x_t$. Hence, tailored to each input batch $x_t$, we compare intermediate features and final box predictions output by a pair of the models $(f_i, f_j)$. To this end, we propose a feature-level similarity function $S_{\text{feat}}(\cdot; \cdot)$ and an output-level box set similarity function $S_{\text{box}}(\cdot; \cdot)$ to effectively quantify the discrepancy between any two 3D detection models (will be discussed in next section). By substituting the inner product with the proposed similarity functions, the original Gram matrix $\mathbf{G}$ evolves into a generalized version $\tilde{\mathbf{G}}$, specially designed for 3D detectors. By combing Eq. (3) and Eq. (4), we can calculate the $\tilde{\mathbf{G}}$ and the optimal synergy weights $\tilde{\mathbf{w}}$ as:

$$\tilde{\mathbf{w}} = (\mathbf{F}^T \mathbf{F})^{-1} \mathbb{1}^K = \tilde{\mathbf{G}}^{-1} \mathbb{1}^K, \ \tilde{\mathbf{G}} = \left( S_{\text{box}} \langle f_i, f_j \rangle \times S_{\text{feat}} \langle f_i, f_j \rangle \right)_{i,j=1}^K, \tag{5}$$

where each element represents the feature and prediction similarity between each pair of historical 3D detection models $(f_i, f_j)$ in $\mathbf{F}$. Next, the super model $f_t^*$ is linearly aggregated by all historical checkpoints $\mathbf{F} \in \mathbb{R}^K$ weighted by $\tilde{\mathbf{w}} \in \mathbb{R}^K$ as follows:

$$f_t^* = \sum_{i=1}^K w_i f_i, \ w_i \in \tilde{\mathbf{w}}, f_i \in \mathbf{F}, \tag{6}$$

where the synergized **super model** $f_t^*$ is used to guide supervision of the current model $f_t$. As illustrated in Fig. 4, for the test sample at time $t$, assembled $f_t^*$ generates the stable pseudo labels $\hat{\mathbf{B}}^t$ to train $f_{t-1} \rightarrow f_t$ with a single iteration, as below:

$$\hat{\mathbf{B}}^t \leftarrow f_t^*(x_t), \ f_t \xleftarrow{\text{train}} \text{aug}(x_t, \hat{\mathbf{B}}^t), \tag{7}$$

where $\text{aug}(\cdot, \cdot)$ indicates data augmentation applied to the pseudo-labeled target point clouds. Recall Eq. (5), the key to finding the optimal synergy weights is to accurately measure the similarities of the features and outputs between a pair of 3D detection models. For each test batch $x_t$, we utilize the intermediate feature map $\mathbf{z}_t$ and the final box prediction set $\mathbf{B}^t$ output by modern 3D detectors (Yin et al., 2021; Lang et al., 2019; Qian et al., 2022; Mao et al., 2023) for similarity measurement. We introduce a feature-level similarity function $S_{\text{feat}}(\cdot; \cdot)$ to assess feature independence between intermediate feature maps $(\mathbf{z}_i, \mathbf{z}_j)$, and an output-level similarity function $S_{\text{out}}(\cdot; \cdot)$ to gauge the discrepancy in box predictions $(\mathbf{B}^i, \mathbf{B}^j)$, where $(\mathbf{z}_i, \mathbf{z}_j)$ and $(\mathbf{B}^i, \mathbf{B}^j)$ are generated by any pair of models $(f_i, f_j)$ within the model bank $\mathbf{F}$.

### 3.1.3 FEATURE-LEVEL SIMILARITY $S_{\text{FEAT}}$

To assess the similarity between feature maps $(\mathbf{z}_i, \mathbf{z}_j)$, we utilize the rank of the feature matrix to determine linear independencies among the feature vectors in $\mathbf{z}_i$ relative to those in $\mathbf{z}_j$. Each single feature vector in the feature map, encodes the information of a small receptive field within the 3D scene. When inferring the input batch $x_t$, if a feature vector in $\mathbf{z}_i$ is linearly independent from the other in $\mathbf{z}_j$, it means $\mathbf{z}_i$ and $\mathbf{z}_j$ are either capturing information from distinct receptive fields, or focusing the same region but interpreting differently. To calculate such independencies, we concatenate $\mathbf{z}_i \oplus \mathbf{z}_j$ and calculate its $\text{rank}(\cdot)$, which identifies the maximum number of linearly independent feature vectors. A *higher* rank of $\mathbf{z}_i \oplus \mathbf{z}_j$ signifies a *greater diversity* of features, because *fewer* feature vectors can be linearly combined by others. To accelerate the computation, we approximate the $\text{rank}(\cdot)$ by computing the nuclear norm of the feature matrix (Recht et al., 2010; Kang et al., 2015). With the input batch $x_t$, the feature-level similarity $S_{\text{feat}}$ between a pair of detection models $(f_i, f_j)$, can be determined as follows:

$$S_{\text{feat}} \langle f_i, f_j \rangle = 1 - \frac{\text{rank}(\mathbf{z}_i \oplus \mathbf{z}_j)}{D}, \ \mathbf{z}_i, \mathbf{z}_j \in \mathbb{R}^{2HW \times D}, \tag{8}$$

where $\mathbf{z}_i, \mathbf{z}_j$ are generated by $f_i$ and $f_j$, respectively. $H$, $W$, $D$ denote the height, width, dimension of the feature map, and $D$ is the maximum possible value of $\text{rank}(\cdot)$. $S_{\text{feat}}$ falls within the range from 0 to 1, and a *higher* value of $S_{\text{feat}}$ indicates *less* feature independence between $\mathbf{z}_i$ and $\mathbf{z}_j$, resulting in *more similar* feature maps yielded by models $f_i$ and $f_j$. Appendix A.1.1 provides empirical evidence and an in-depth analysis of the rank-based similarity metric.

### 3.1.4 Output-level Box Similarity $S_{BOX}$

To assess the similarity between box predictions $(\mathbf{B}^i, \mathbf{B}^j)$, we treat it as a *set matching* problem, since $\mathbf{B}^i$ and $\mathbf{B}^j$ represent sets of predicted bounding boxes that can vary in size. To solve it, we apply the Hungarian matching algorithm (Carion et al., 2020; Stewart et al., 2016), a robust method for bipartite matching that ensures the best possible one-to-one correspondence between two sets of box predictions. We extend the smaller box set with $\emptyset$ to equalize the sizes of both sets to $N$. To determine an optimal bipartite matching between these two equal-sized sets, the Hungarian algorithm is used to find a permutation of $N$ elements: $p \in \mathbf{P}$, which yields the lowest cost:

$$\tilde{p} = \arg\min_{p \in \mathcal{P}} \sum_n \mathcal{L}_{\text{box}}(\mathbf{B}_n^i; \mathbf{B}_{p(n)}^j), \tag{9}$$

where $\mathbf{B}_n^i$ is $n$-th box in the set $\mathbf{B}^i$ predicted by the model $f_i$. The box loss $\mathcal{L}_{\text{box}}(\cdot; \cdot)$ is calculated by the intersection-over-union (IoU) plus the L1 distance covering the central coordinates, dimensions, and orientation angles between a pair of boxes: $\mathbf{B}_n^i$ and its corresponding box indexed by $p(n)$. The indicator function $\mathbb{1}_{\{c_n^i \neq \emptyset\}}$ means the cost is calculated only when the category of $n$-th object is not the padded $\emptyset$. The next step is to compute the total Hungarian loss (Carion et al., 2020) for all pairs of matched boxes:

$$S_{\text{box}}(f_i, f_j) = \left( \sum_{n=1}^N \mathbb{1}_{\{c_n^i \neq \emptyset\}} \mathcal{L}_{\text{box}}(\mathbf{B}_n^i; \mathbf{B}_{\tilde{p}(n)}^j) \right)^{-1}, \tag{10}$$

where $\tilde{p}$ is the optimal assignment computed in the Eq. (9). We normalize the result to the same range as $S_{\text{feat}}$ using a sigmoid function. A *higher* value of $S_{\text{box}}$ indicates that models $f_i$ and $f_j$ predict *more overlapping* boxes.

### 3.2 Model Bank Update (Phase 3)

In TTA-3OD, with the increasing number of test batches, simply inserting checkpoints at all timestamps to the model bank $\mathbf{F}$ results in substantial memory cost and knowledge redundancy. As observed from our empirical results (Fig. 6), assembling the latest 20 checkpoints yields similar performance to assembling only the latest 5 checkpoints. Hence, our goal is to make the $\mathbf{F}$ retain only a minimal yet highly diverse set of checkpoints. To this end, we regularly update the model bank, by adding new models and removing redundant ones every time $L$ online batches are inferred. As illustrated in Fig. 4, upon adapting every $L$ samples $\{x_l, x_{l+1}, \cdots, x_L\}$, a single checkpoint $f_i \in \mathbf{F}$ engages in the assembly $L$ times, thereby $L$ synergy weights are calculated and stored into a list $\{w_{(l,i)}, w_{(l+1,i)}, \cdots, w_{(L,i)}\}$. Given $K$ checkpoints in $\mathbf{F}$, a synergy matrix is defined as $\mathbf{W} = (w_{(l,k)})_{l=1,k=1,}^{L,K} \in \mathbb{R}^{L \times K}$. After inferring $L$ batches, we average across the $L$ dimension to calculate the mean synergy weight for each checkpoint, denoted as:

$$\bar{\mathbf{w}} = \{\bar{w}_1, \cdots, \bar{w}_K\} \in \mathbb{R}^K. \tag{11}$$

Finally, we remove the model with **the lowest** mean synergy weight in $\bar{\mathbf{w}}$ and add the current model $f_t$ to the bank as follows:

$$\mathbf{F} \leftarrow \mathbf{F} \setminus \{f_i\}, \ i = \texttt{index}(\bar{\mathbf{w}}; \texttt{min}(\bar{\mathbf{w}})), \ \mathbf{F} \leftarrow \mathbf{F} \cup \{f_t\}, \tag{12}$$

where $i$ denotes the index of the minimum mean synergy weight in $\bar{\mathbf{w}}$. By updating the model bank this way, we maintain a fixed number of $K$ checkpoints in $\mathbf{F}$, each carrying unique insights.

## 4 Experiments

### 4.1 Experimental Setup

**Datasets and TTA-3OD Tasks.** We perform extensive experiments across three widely used LiDAR-based 3D object detection datasets: **KITTI** (Geiger et al., 2012), **Waymo** (Sun et al., 2020), and **nuScenes** (Caesar et al., 2020), along with a recently introduced dataset simulating real-world corruptions, **KITTI-C** (Kong et al., 2023) for TTA-3OD challenges. We firstly follow (Yang et al., 2021; 2022; Chen et al., 2023a) to tackle cross-dataset test-time adaptation tasks (*e.g.* Waymo → KITTI and nuScenes → KITTI-C), which include adaptation (i) across object shifts (*e.g.*, scale and point density), and (ii) across environmental shifts (*e.g.*, deployment locations and LiDAR beams). We also conduct experiments to tackle a wide array of real-world corruptions (*e.g.*, KITTI → KITTI-C) covering: Fog, Wet Conditions (Wet.), Snow, Motion blur (Moti.), Missing beams (Beam.),

Table 1: Results of test-time adaptation for 3D object detection across different datasets. We report $AP_{BEV}$ / $AP_{3D}$ in moderate difficulty. Oracle means fully supervised training on the target dataset. We indicate the best adaptation result by **bold**.

| Method | Venue | TTA | Waymo →KITTI | | nuScenes →KITTI | |
|---|---|---|---|---|---|---|
| | | | $AP_{BEV}$ / $AP_{3D}$ | Closed Gap | $AP_{BEV}$ / $AP_{3D}$ | Closed Gap |
| No Adapt. | - | - | 67.64 / 27.48 | - | 51.84 / 17.92 | - |
| SN | CVPR'20 | ✗ | 78.96 / 59.20 | +72.33% / +69.00% | 40.03 / 21.23 | +37.55% / +5.96% |
| ST3D | CVPR'21 | ✗ | 82.19 / 61.83 | +92.97% / +74.72% | 75.94 / 54.13 | +76.63% / +65.21% |
| Oracle | - | - | 83.29 / 73.45 | - | 83.29 / 73.45 | - |
| Tent | ICLR'21 | ✓ | 65.09 / 30.12 | −16.29% / +5.74% | 46.90 / 18.83 | −15.71% / +1.64% |
| CoTTA | CVPR'22 | ✓ | 67.46 / 35.34 | −1.15% / +17.10% | 68.81 / 47.61 | +53.96% / +53.47% |
| SAR | ICLR'23 | ✓ | 65.81 / 30.39 | −11.69% / +6.33% | 61.34 / 35.74 | +30.21% / +32.09% |
| MemCLR | WACV'23 | ✓ | 65.61 / 29.83 | −12.97% / +5.11% | 61.47 / 35.76 | +30.62% / +32.13% |
| **MOS** | - | ✓ | **81.90 / 64.16** | **+91.12% / +79.79%** | **71.13 / 51.11** | **+61.33% / +59.78%** |

Table 2: Results of test-time adaptation for 3D object detection across different corruptions (KITTI → KITTI-C) at heavy level. $AP_{3D}$ at easy/moderate/hard difficulty of KITTI metric are reported.

| | No Adaptation | Tent | CoTTA | SAR | MemCLR | MOS |
|---|---|---|---|---|---|---|
| Fog | 84.66/72.85/68.23 | 85.11/73.02/68.73 | 85.10/72.94/68.49 | 84.64/ 72.52/68.14 | 84.71/72.70/68.23 | **85.22/74.02/69.11** |
| Wet. | 88.23/78.82/76.25 | 88.13/78.79/76.36 | 88.27/79.02/76.43 | 88.11/78.60/76.23 | 88.04/78.68/76.25 | **89.32/81.91/77.79** |
| Snow | 71.11/63.92/59.07 | 71.71/64.48/59.50 | 71.83/64.59/59.45 | 72.29/64.08/58.78 | 72.33/63.84/58.74 | **75.07/67.87/62.72** |
| Moti. | 43.35/39.23/38.21 | 43.02/39.15/38.15 | 44.48/39.68/38.62 | 42.89/39.13/38.12 | 42.86/38.44/37.57 | **44.70/41.63/40.59** |
| Beam. | 75.42/57.49/53.93 | 76.62/58.01/53.85 | 76.39/57.50/53.98 | 75.98/57.67/53.75 | 75.70/57.48/53.49 | **78.74/60.48/55.91** |
| CrossT. | 87.98/78.46/75.49 | 87.62/77.92/74.67 | 86.78/76.05/72.22 | 87.74/77.59/74.51 | 86.20/77.11/74.25 | **89.22/79.10/**75.47 |
| Inc. | 46.98/30.33/25.68 | 47.82/30.87/26.44 | 49.92/32.77/27.85 | 47.27/30.46/26.42 | 49.99/32.08/27.47 | **59.13/40.29/34.53** |
| CrossS. | 68.44/47.37/41.08 | 68.89/46.75/41.17 | 69.15/47.11/40.80 | 68.17/46.45/40.63 | 69.17/47.28/40.90 | **71.87/49.28/43.68** |
| Mean | 70.77/58.56/54.74 | 71.11/58.62/54.86 | 71.49/58.71/54.73 | 70.89/58.31/54.57 | 71.13/58.45/54.61 | **74.16/61.82/57.48** |

Crosstalk (Cross.T), Incomplete echoes (Inc.), and Cross-sensor (Cross.S). Finally, we address the hybrid shift cross-corruptions (*e.g.*, Waymo → KITTI-C), where cross-dataset inconsistency and corruptions coexist in test scenes. The implementation details are included in Appendix A.4.

**Baseline Methods.** We compare the proposed method using voxel-based backbone (*e.g.*, SECOND) with a diverse range of baseline methods: (i) **No Adapt.** refers to the model pretrained from the source domain, directly infer the target data; (ii) **SN** (Wang et al., 2020b) is a *weakly supervised domain adaptation* method for 3D detection by rescaling source object sizes with target statistics for training; (iii) **ST3D** (Yang et al., 2021) is the pioneering *unsupervised domain adaption* method for 3D detection with multi-epoch pseudo labeling-based self-training; (iv) **Tent** (Wang et al., 2021) is an *online TTA* method which optimizes the model by minimizing entropy of its prediction; (v) **CoTTA** (Wang et al., 2022a) is an *online TTA* approach which employs mean-teacher to provide supervision with augmentations and restores neurons to preserve knowledge; (vi) **SAR** (Niu et al., 2023) improves Tent by sharpness-aware and reliable entropy minimization for *online TTA*; (viii) **MemCLR** (VS et al., 2023) is the first work for *online adaptation on image-based object detection*, by utilizing mean-teacher to align the instance-level features with a memory module; (ix) **Oracle** means a *fully supervised* model trained on the target domain.

## 4.2 MAIN RESULTS AND ANALYSIS

We present and analyze the main results of car detection using SECOND (Yan et al., 2018) as the backbone 3D detector, when comparing the proposed MOS against baseline methods across three types of domain shifts. The results of multiple object classes are detailed in the Appendix A.5, and the examination of alternative backbone detectors (*e.g.*, PVRCNN (Shi et al., 2020) and DSVT (Wang et al., 2023a)), is presented in the Appendix A.7. Comparisons with linear and non-linear ensemble methods are included in Appendix A.1.4.

**Cross-dataset Shifts.** We conducted comprehensive experiments on two cross-dataset 3D adaptation tasks, as reported in Tab. 1, when compared to the best TTA baseline CoTTA (Wang et al., 2022a), the proposed MOS consistently improves the performance in Waymo → KITTI and nuScenes → KITTI tasks by a large margin of 81.5 % and 7.4% in $AP_{3D}$, respectively. This improvement significantly closes the performance gap (91.12% and 79.79%) between No Adapt. and Oracle. Remarkably, the proposed MOS outperforms the UDA method ST3D (Yang et al., 2021), which requires multi-epoch training: MOS's 64.16% vs. ST3D's 61.83% in $AP_{3D}$, and even demonstrates comparable performance to the upper bound performance (Oracle): MOS's 81.90% vs. Oracle's 83.29% in $AP_{BEV}$ when adapting from Waymo to KITTI. The results and analysis of additional cross-dataset transfer tasks (Waymo → nuScenes) are provided in Appendix A.2.

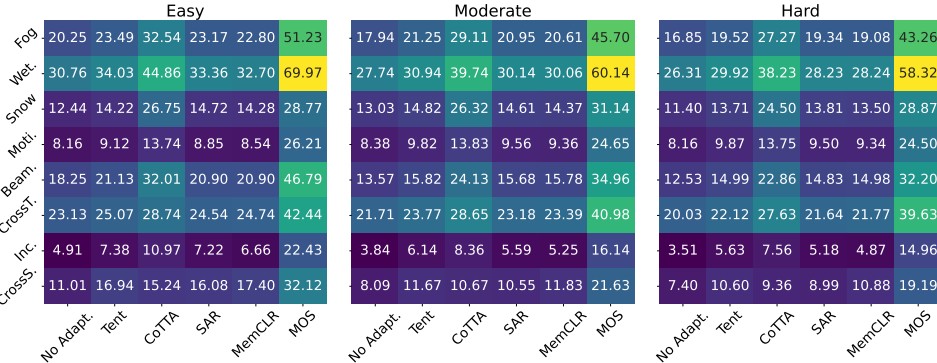

Figure 5: Heatmap intuitively presenting the results of TTA-3OD across hybrid cross-corruption shifts (Waymo → KITTI-C) in heavy difficulty. Darker/lighter shades indicate lower/higher performance.

**Corruption Shifts.** We carry out comprehensive experiments (KITTI → KITTI-C) across eight different types of real-world corruptions to validate the effectiveness of the proposed MOS to mitigate these corruption shifts. As reported in Tab. 2, MOS surpasses all TTA baselines across every type of corruption, achieving $AP_{3D}$ gains of 3.73%, 5.56%, and 4.78% at easy, moderate, and hard levels, respectively. Notably, MOS greatly addresses the most severe shift caused by incomplete echoes, improving 23.98% at the hard difficulty level. These promising results highlight MOS's robustness to adapt 3D models to various corrupted scenarios, particularly those challenging ones.

**Cross-corruption Shifts.** To address the hybrid shifts where cross-dataset discrepancies and corruptions simultaneously occur, we conduct experiments to adapt 3D detectors from Waymo to KITTI-C. We visualize the results utilizing the heatmap (in Fig. 5). It is observed that the shades in the last column (MOS), are significantly lighter than those in all other columns (TTA baselines),

Table 3: Ablative study of the proposed MOS on Waymo → KITTI. The best result is highlighted in **bold**.

| Ablation | $AP_{3D}$ | | | $AP_{BEV}$ | | |
|---|---|---|---|---|---|---|
| | Easy | Moderate | Hard | Easy | Moderate | Hard |
| w/o Ensemble | 45.17 | 43.14 | 41.48 | 71.36 | 68.17 | 68.96 |
| Mean Ensemble | 48.56 | 45.89 | 44.37 | 72.72 | 70.34 | 70.15 |
| MOS w/o $S_{feat}$ | 67.02 | 59.98 | 58.76 | 91.85 | 81.90 | 79.77 |
| MOS w/o $S_{box}$ | 72.99 | 61.40 | 60.09 | 88.52 | 80.15 | 79.06 |
| MOS | **74.09** | **64.16** | **62.33** | **91.85** | **81.90** | **81.78** |

indicating the superior performance of MOS over all other methods. It is worth noting that, the results without any adaptation (column 1) remarkably degrade, due to the challenging hybrid shifts, for example, only 3.51% for incomplete echoes and 7.40% for cross-sensor at a hard level. Our approach greatly enhances adaptation performance for these two challenging corruptions by **97.99%** and **76.38%**, respectively, compared to the best baselines. Thus, prior TTA baselines fail to mitigate the significant domain shifts across 3D scenes, whereas the proposed MOS successfully handles various cross-corruption shifts, surpassing the highest baseline and direct inference by **67.3%** and **161.5%**, respectively, across all corruptions.

### 4.3 ABLATION AND SENSITIVITY STUDY

**Ablation Study.** To validate the effectiveness of each component in the proposed MOS, we conduct ablation experiments on Waymo → KITTI and report the $AP_{3D}$ across three difficulty levels. First, we remove the pivotal component in our method, model assembly, opting instead to simply use the model at the current timestamp for pseudo-label generation. Tab. 3 shows that without assembly (row 1), the adaptation yields suboptimal results, achieving only 43.14% in moderate $AP_{3D}$. Then we assemble the recent five checkpoints by averaging their parameters (row 2), and observe a marginal improvement in moderate $AP_{3D}$ to 45.89%. While our proposed assembly strategy (row 5), identifying and assembling the most useful 5 checkpoints through weighted averaging, significantly outperforms average aggregation by 39.8%. Next, we remove each of the proposed similarity functions used to calculate the synergy weights for the model ensemble. As shown in Tab. 3, removing either $S_{feat}(\cdot;\cdot)$ or $S_{box}(\cdot;\cdot)$ leads to a notable decrease in moderate $AP_{3D}$ by 6.9% and 4.5%, respectively, proving that both functions accurately gauge similarity between any pair of 3D detection models. Additional ablation studies can be found in Appendix A.1.

**Sensitivity to Hyperparameters.** In this section, we examine how the adaptation performance of the proposed MOS is affected by the hyperparameters $L$ and $K$, which determines the period of the model bank update(*i.e.*, every $L$ batches of test data) and the bank capacity. To study $L$, we conduct experiments with values ranging from 64 to 128. As illustrated in the Fig. 6 (*left*), the $AP_{BEV}$ (blue

curve) remains stable, while the $AP_{3D}$ (green curve) increases as $L$ grows from 64 to 96. This occurs because a larger $L$ reduces the model update frequency, enabling the model bank to retain older checkpoints that hold distinct long-term knowledge. Upon reaching 96, $AP_{3D}$ becomes stable with a slight fluctuation of 1.67%.

To assess the informativeness and significance of the $K$ checkpoints selected by the proposed MOS, we developed a comparative analysis between the standard MOS and a simplified variant: *latest-first*

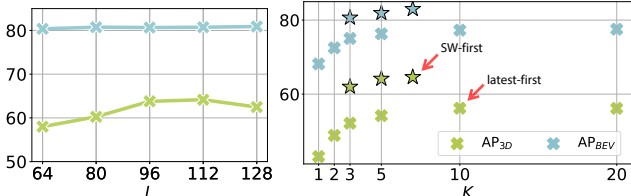

MOS. This variant (crosses in Fig. 6 *right*) utilizes the most recent $K$ checkpoints and employs the same ensemble strategy as MOS (*i.e.*, Eq. (5) and Eq. (6)). The standard MOS, that maintains $K$ checkpoints with high synergy weights (SW) (Sec. 3.2), is referred to as SW-first MOS (stars in Fig. 6 *right*). As shown in Fig. 6, latest-first MOS presents an obvious improvement in $AP_{3D}$ from 43.14% to 54.18% and in $AP_{BEV}$ from 68.17% to 76.29% when assembling more latest

Figure 6: *Left*: Sensitivity to the period of model bank update $L$. *Right*: Comparison between two MOS-based ensemble strategies (SW-first vs. latest-first) with increasing model bank size $K$. Both $AP_{BEV}$ and $AP_{3D}$ are plotted when adapting the 3D detector (Yan et al., 2018) from Waymo to KITTI.

checkpoints ($K$ increases from 1 to 5). However, expanding the ensemble to 20 models yields a marginal increase to 56.16% in $AP_{3D}$ and 77.54% in $AP_{BEV}$, with a substantial memory cost. The SW-first MOS, which dynamically utilizes only 3 historical checkpoints based on averaged SW, outperforms latest-first MOS with 20 checkpoints by 18.42% in $AP_{3D}$ and 5.84% in $AP_{BEV}$, meanwhile, reducing memory usage by 85%. This efficiency highlights our method's ability to selectively keep the most pivotal historical models in the model bank, optimizing both detection performance and memory expenditure.

## 4.4 COMPLEXITY ANALYSIS

This section analyzes the complexity of the proposed MOS. Recall that $K$ models in the bank $\mathbf{F}$ infer every test batch $x_t$ from the datasets $\{x_t\}_{t=1}^{T}$, the theoretical time complexity is asymptotically equivalent to $\mathcal{O}(T)$, as $K$ is a constant. Concerning the space complexity, we assume that a 3D detection network requires $N_{param}$ parameters for memory storage, and the space complexity is bounded by $\mathcal{O}(N_{param})$. Theoretically, the proposed MOS exhibits a manageable linear complexity growth with respect to model size and dataset size.

**Time and Memory Usage** We report the adaptation speed (seconds per frame) and GPU memory usage (MiB) of implemented methods in Fig. 7. It is observed that the proposed MOS significantly outperforms the top baseline, improving $AP_{3D}$ by 21.4% while only requiring an additional 0.255 seconds per frame compared to the baseline average. In terms of memory usage, mean-teacher frameworks such as Mem-CLR (19,087 MiB) and CoTTA (15,099 MiB) consume more resources as they simultaneously

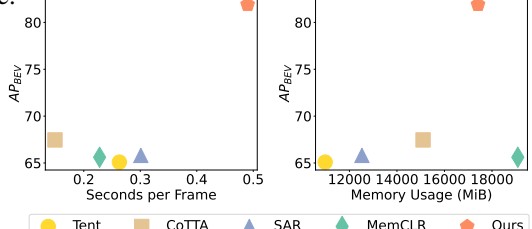

Figure 7: Time and memory usage of method implementations when adapting the 3D detector (Yan et al., 2018) from Waymo to KITTI dataset.

load dual networks in the GPU. The substantial consumption of MemCLR mainly stems from its extra transformer-based memory module. Compared to MemCLR, our MOS maintains a lower memory footprint at 17,411 MiB. Overall, MOS offers substantially superior adaptation performance with a manageable increase in memory and time consumption.

## 5 CONCLUSION

This paper is an early attempt to explore the test-time adaptation for LiDAR-based 3D object detection and introduces an effective model synergy approach that leverages long-term knowledge from historical checkpoints to tackle this task. Comprehensive experiments confirm that MOS efficiently adapts both voxel- and point-based 3D detectors to various scenes in a single pass, addressing real-world domain shifts sourced from cross-dataset, corruptions, and cross-corruptions. The main limitation is its high computational demand and future efforts will focus on improving the time and space efficiency of the algorithm, through storing, and assembly of only a small portion of model parameters (*e.g.*, selected layers or modules) to facilitate faster adaptation at test time.

ACKNOWLEDGMENT

This work was supported by the Australian Research Council (DE240100105, DP240101814, DP230101196).

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

# A APPENDIX

This Appendix provides additional details, experiments, and analysis of the proposed MOS, summarized as follows:

- **Algorithm. 1:** Detailed algorithm of the proposed MOS framework.
- **Sec. A.1:** Additional ablation and component study on a more challenge transfer tasks: Waymo $\rightarrow$ KITTI-C.
    - Study on similarity measurement methods (A.1.1)
    - Study on feature selection for similarity comparison (A.1.2)
    - Study on MOS's ability to retain long-term knowledge (A.1.3)
    - Study on direct use of trivial ensemble method (A.1.4)
- **Sec. A.2:** Experimental results and analysis on additional transfer task: Waymo $\rightarrow$ nuScenes.
- **Sec. A.3:** Discussion about related model ensemble methods.
- **Sec. A.4:** Implementation details on computing devices, hyperparameter selection, data augmentations, pseudo-labeling, baseline losses and ensemble techniques in MOS.
- **Sec. A.5:** Experimental results and analysis across multiple classes to address both cross-dataset and cross-corruption shifts.
- **Sec. A.6:** Visualizaion of 3D box predictions by checkpoints with different synergy weights.
- **Sec. A.7:** Experimental results with different backbone 3D detectors.
- **Sec. A.8:** Qualitative study with visualized 3D box predictions from different methods.

## A.1 ADDITIONAL ABLATION/COMPONENT STUDY

To further validate the effectiveness of each component in the proposed MOS, we conducted an ablation study on a more challenging transfer task: Waymo $\rightarrow$ KITTI-C under the corruption "incomplete echo". As reported in Tab. 4, we observed a significant performance drop when either feature similarity or box similarity was removed: 25.4% and 17.2% reductions in $AP_{3D}$ at hard difficulty, respectively. Compared to the 5.7% and 3.6% drop in the Waymo $\rightarrow$ KITTI task, MOS proves to be especially effective in more difficult cross-corruption transfer tasks.

Table 4: Ablative study of the proposed MOS on Waymo $\rightarrow$ KITTI-C, under the corruption "**incomplete echo**". MOS cosine $S_{feat}$ refers to the use of cosine similarity instead of rank-based similarity. MOS RoI $S_{feat}$ denotes that RoI features from the second stage, rather than BEV maps, are used by MOS. The best result is highlighted in **bold**.

| | $AP_{3D}$ | | | $AP_{BEV}$ | | |
|---|---|---|---|---|---|---|
| Ablation | Easy | Moderate | Hard | Easy | Moderate | Hard |
| MOS w/o $S_{feat}$ | 18.79 | 13.68 | 11.93 | 42.62 | 29.34 | 26.30 |
| MOS w/o $S_{box}$ | 20.13 | 14.39 | 12.38 | 44.34 | 30.77 | 27.77 |
| MOS cosine $S_{feat}$ | 17.74 | 12.51 | 11.04 | 42.20 | 29.01 | 26.18 |
| MOS RoI $S_{feat}$ | 19.13 | 13.73 | 11.84 | 44.14 | 30.72 | 27.71 |
| MOS | **22.43** | **16.14** | **14.96** | **44.18** | **32.05** | **29.58** |

### A.1.1 RANK-BASED SIMILARITY VS. COSINE SIMILARITY

In this section, we discuss the advantages of the rank-based similarity method and its superiority over the conventional cosine similarity. The key advantage is its ability to capture feature similarity not only between a pair of point clouds but also **within each single point cloud**. In our method, we concatenate $\mathbf{z}_i$ and $\mathbf{z}_j$ and compute the rank of the concatenated matrix. High similarity (**low rank value**) occurs in two cases: (1) most features in $\mathbf{z}_i$ are linearly dependent on features in $\mathbf{z}_j$, indicating that $\mathbf{z}_i$ and $\mathbf{z}_j$ are similar, or (2) even if $\mathbf{z}_i$ and $\mathbf{z}_j$ are very different, the features within $\mathbf{z}_i$ (or within $\mathbf{z}_j$) are linearly dependent on each other, for example, a point cloud contains cars only with very similar shapes and sizes. In this case, the concatenated feature will still produce a low rank value due to the absence of many diverse features although concatenated, indicating **redundancy**. In contrast, cosine similarity is based purely on the comparison between a pair of frames and ignores feature similarity within each point cloud. To support our claim, we provide experiments in Tab. 4. It was observed that utilizing cosine similarity (row 3) results in a significant performance **drop of 29%** in $AP_{3D}$ Moderate compared to rank-based similarity measurements (row 5). Furthermore, when comparing it to MOS without feature similarity (row 1), there is no performance gain. These findings

underscore the **importance of feature diversity within point clouds**, which cosine similarity **fails** to capture.

### A.1.2 SELECTION OF INTERMEDIATE FEATURES

In our method, we empirically utilize the **BEV feature map** of size 128 * 128 * 256 from the last layer of the encoder in both SECOND and PV-RCNN, as it is directly used to generate bounding boxes thus containing global fine-grained information about the point clouds. We also conduct experiments on extracted region of interest (RoI) features (*i.e.*, 100 RoI features of dimension 128). As shown in Tab. 4, using RoI features (row 4) yields lower performance (**3.1%** in $AP_{3D}$ Hard) than the BEV map (row 5). This occurs as RoI features focus only on foreground objects, resulting in information loss in environments.

### A.1.3 STUDY ON LONG-TERM KNOWLADGE

We further investigate the capability of MOS in selecting and preserving checkpoints that contain **long-term knowledge**. Specifically, we perform experiments where the final stored checkpoints, after completing the test-time adaptation, are ensembled into the super model to infer the early data (*i.e.*, the first 32 point clouds). The results in Tab. 5 confirm that even after running for a long time and reaching the end of the target datasets, the final checkpoints maintain strong performance on the initial 32 point clouds, significantly outperforming the 32nd checkpoint by 239% in $AP_{3D}$. Furthermore, the high recall rate of 69.34 demonstrates that nearly all objects in the early batches are detected, highlighting the capacity of the saved checkpoints to retain diverse and long-term knowledge.

Table 5: Performance of the 32th checkpoint and final stored checkpoints of MOS on the Early Set (*i.e.*, the first 32 point clouds), when adapting from Waymo to KITTI-C, under the corruption "**incomplete echo**". Recall@0.5 denotes the recall metric calculated at an IoU of 0.5.

| MOS | $AP_{BEV}$ | $AP_{3D}$ | Recall@0.5 |
|---|---|---|---|
| MOS (32th Checkpoint) on Early Set | 15.03 | 4.14 | 64.04 |
| MOS (Final Checkpoints) on Early Set | 33.38 | 14.04 | 69.34 |

### A.1.4 COMPARARION WITH TRIVIAL ENSEMBLE METHODS

**Least Squares.** To investigate whether trivial ensemble methods can effectively solve for the optimal synthetic model in the TTA-3OD task, we employed the simplest approach: least squares, to ensemble only a single layer of 3D detection model. Specifically, we treat the frozen encoder as producing a fixed feature map $z$, and we apply least squares (LS) to the final layer, which consists of only 256 parameters. As the results clearly demonstrate in Tab. 6, applying LS to a single layer with very few parameters (*i.e.*, 256) of a 3D detection model failed to address the TTA scenarios, yielding a notably lower-bound accuracy of 4.93 in $AP_{3D}$.

Table 6: Performance comparison between trivial ensemble methods and the proposed MOS in TTA-3OD, when adapting from Waymo to KITTI-C, under the corruption "**incomplete echo**". AP at a moderate difficulty level is reported.

| | $AP_{BEV}$ | $AP_{3D}$ |
|---|---|---|
| No Adapt | 15.15 | 3.77 |
| Least Squares (Last Layer) | 18.28 | 4.93 |
| Bagging (3 models) | 27.59 | 14.30 |
| **MOS** | **32.05** | **16.14** |

**Bagging.** We also explore a traditional **non-linear** ensemble method, bagging (Breiman, 1996). This method involves training multiple models on randomly sampled subsets of the training set, which makes it inapplicable for the TTA-3OD task. In our implementation, for each test batch, we randomly select one model from three to update the weights, followed by non-maximum suppression (NMS) with a threshold of 0.7 to obtain the final predictions from the box outputs of all three models. Our empirical results show that bagging consumes 34,952 MiB of GPU memory due to the simultaneous loading of three 3D detectors, which is significantly higher than the proposed MOS method (17,411 MiB). Despite the heavy GPU usage, bagging is outperformed by the proposed MOS method, with a 11.4% lower $AP_{3D}$, indicating that non-linear ensemble methods are computationally expensive and less effective, making them unsuitable for the TTA-3OD task.

Table 8: [Complete Results] Cross-dataset results in moderate difficulty across multiple classes. We report average precision (AP) for bird's-eye view ($AP_{BEV}$) / 3D ($AP_{3D}$) of **car**, **pedestrian**, and **cyclist** with IoU threshold set to 0.7, 0.5, and 0.5 respectively. We indicate the best adaptation result by **bold**, and the highlighted row represents the proposed method.

| Task | Method | Car | Pedestrian | Cyclist |
|---|---|---|---|---|
| Waymo → KITTI | No Adapt. | 67.64 / 27.48 | 46.29 / 43.13 | 48.61 / 43.84 |
| | SN (Wang et al., 2020b) | 78.96 / 59.20 | 53.72 / 50.44 | 44.61 / 41.43 |
| | ST3D (Yang et al., 2021) | 82.19 / 61.83 | 52.92 / 48.33 | 53.73 / 46.09 |
| | Oracle | 83.29 / 73.45 | 46.64 / 41.33 | 62.92 / 60.32 |
| | Tent (Wang et al., 2021) | 65.09 / 30.12 | 46.21 / 43.24 | 46.59 / 41.72 |
| | CoTTA (Wang et al., 2022a) | 67.46 / 35.34 | 49.02 / 45.03 | **55.31** / 50.07 |
| | SAR (Niu et al., 2023) | 65.81 / 30.39 | 47.99 / 44.72 | 47.14 / 41.17 |
| | MemCLR (VS et al., 2023) | 65.61 / 29.83 | 48.25 / 44.84 | 47.49 / 41.67 |
| | **MOS** | **81.90 / 64.16** | **50.25 / 46.51** | 54.12 / **51.36** |
| nuScenes → KITTI | No Adapt. | 51.84 / 17.92 | 39.95 / 34.57 | 17.61 / 11.17 |
| | SN (Wang et al., 2020b) | 40.03 / 21.23 | 38.91 / 34.36 | 11.11 / 5.67 |
| | ST3D (Yang et al., 2021) | 75.94 / 54.13 | 44.00 / 42.60 | 29.58 / 21.21 |
| | Oracle | 83.29 / 73.45 | 46.64 / 41.33 | 62.92 / 60.32 |
| | Tent (Wang et al., 2021) | 46.90 / 18.83 | 39.69 / 35.53 | 19.64 / 10.49 |
| | CoTTA (Wang et al., 2022a) | 68.81 / 47.61 | 40.90 / 36.64 | 20.68 / 13.81 |
| | SAR (Niu et al., 2023) | 65.81 / 30.39 | 39.78 / 35.63 | 19.56 / 10.26 |
| | MemCLR (VS et al., 2023) | 65.61 / 29.83 | 40.31 / 35.14 | 19.21 / 9.66 |
| | **MOS** | **71.13 / 51.11** | **42.94 / 38.45** | **21.65 / 18.16** |

## A.2 ADDITIONAL TRANSFER TASK

To further assess the effectiveness of the proposed MOS, we perform an additional challenging transfer task: from Waymo to NuScenes, with the $AP_{3D}$ results presented in Tab. 7. Notably, ST3D (Yang et al., 2021), a multi-round UDA method, shows **limited** performance (*i.e.*, 20.19), likely due to the significant domain gap (*e.g.*, differences in beam numbers). Despite this challenge, the proposed MOS surpasses all TTA baselines, achieving state-of-the-art performance.

Table 7: Results ($AP_{3D}$ at moderate level difficulty) of TTA-3OD on adapting Waymo → nuScenes using SECOND.

| No Adapt. | ST3D | Tent | CoTTA | SAR | MemCLR | Ours |
|---|---|---|---|---|---|---|
| 17.24 | 20.19 | 17.31 | 17.35 | 17.74 | 18.22 | **18.75** |

## A.3 DISCUSSION ABOUT RELEVANT MODEL ENSEMBLE METHODS

Gao et al. (2022) utilizes gradient information and parameter space advancements during checkpoint averaging. However, this method is **infeasible** for TTA-3OD due to **noisy** gradients from the lack of ground truth labels. Additionally, computation on gradient is **expensive**, hindering efficient adaptation. Also, this work (Gao et al., 2022) requires optimization on **development data** (val set), which is **unavailable** during test-time adaptation. In contrast, our MOS measures output differences between checkpoint pairs, ensuring diverse knowledge without relying on gradients or development data.

Kaddour (2022) examines EMA, varying the latest checkpoints (ckpt) and saving frequencies. We investigated **the same ensemble strategy using the latest ckpt**, as shown in **Fig. 6** of the main paper. The **"latest-first"** strategy aggregates the most recent K ckpt. Our MOS strategy selecting the 3 **most important** ckpt, outperforms the ensemble of the latest twenty ckpt. This improvement arises because recent ckpt converge, leading to similar parameters and loss of long-term unique knowledge, ultimately failing to generalize to the target domain. Our MOS effectively mitigates this issue.

Matena & Raffel (2022) presents a merging procedure using the Laplace approximation, approximating each model's posterior as a Gaussian distribution with the precision matrix as its Fisher information. This approach is **infeasible** for the TTA-3OD task, as it necessitates per-example data to estimate the Fisher matrix, which is unavailable in online TTA since the test set is not provided in advance.

Table 9: TTA-3OD results (easy/moderate/hard AP$_{3D}$) of **pedestrian** class under the cross-corruption scenario (Waymo → KITTI-C) at heavy corruption level.

| | No Adaptation | Tent (Wang et al., 2021) | CoTTA (Wang et al., 2022a) | SAR (Niu et al., 2023) | MemCLR (VS et al., 2023) | MOS |
|---|---|---|---|---|---|---|
| Fog | 30.48/26.15/23.61 | 31.22/26.68/23.99 | 31.29/26.69/24.05 | 30.68/25.94/23.70 | 30.51/26.02/23.77 | **32.38/27.51/24.85** |
| Wet. | 49.10/44.44/41.74 | 49.13/44.58/41.85 | 49.14/45.01/42.23 | 49.18/44.59/41.97 | 49.09/44.55/41.81 | **51.34/46.88/43.16** |
| Snow | 47.22/42.42/39.19 | 47.55/42.79/39.44 | 46.30/41.62/38.11 | 47.42/42.85/39.54 | 47.68/42.78/39.45 | **49.42/44.82/40.79** |
| Moti. | 27.18/25.02/23.29 | 27.47/25.25/23.43 | 27.28/25.43/23.41 | 27.34/25.15/23.31 | 27.44/25.19/23.36 | **27.54/25.93/23.86** |
| Beam. | 32.47/27.89/25.27 | 34.50/30.55/28.18 | 32.22/27.41/25.13 | **34.83/30.74/28.54** | 34.53/30.30/28.16 | 34.02/29.73/27.35 |
| CrossT. | 47.42/43.08/40.37 | 47.66/43.37/40.51 | 47.76/43.29/40.39 | 48.13/43.65/40.71 | 47.87/43.58/40.48 | **50.78/45.42/41.73** |
| Inc. | 49.28/44.79/42.21 | 49.18/44.80/42.11 | 49.36/45.39/42.77 | 49.22/44.70/42.24 | 49.01/44.76/42.11 | **51.06/46.83/43.31** |
| CrossS. | 22.46/18.40/16.08 | 27.70/22.82/20.30 | 22.11/17.88/15.98 | **27.99**/23.20/21.36 | 27.23/**23.70**/21.63 | 26.74/22.94/20.89 |
| Mean | 38.20/34.00/31.47 | 39.30/35.11/32.48 | 38.18/34.09/31.52 | 39.35/35.10/32.67 | 39.17/35.11/32.60 | **40.41/36.26/33.24** |

Wortsman et al. (2022) supports our findings that averaging weights of models with different hyperparameters can improve accuracy and robustness. However, it doesn't consider model weighting based on checkpoint importance and uniqueness, unlike our proposed MOS. Additionally, the "learned soup" and "greedy soup" methods in paper (Wortsman et al., 2022) require a held-out validation set to compute mixing coefficients, making them **unsuitable** for test-time scenarios.

In summary, all the aforementioned papers focus on **multi-epoch offline training**, potentially requiring labels and validation sets, and neglect the estimation and selection of important ckpt for ensemble. In contrast, our ensemble method is uniquely designed for online Test-Time Adaptation (TTA) in 3D object detection.

### A.4 IMPLEMENTATION DETAILS

Our code is developed on the OpenPCDet (Team, 2020) point cloud detection framework, and operates on a single NVIDIA RTX A6000 GPU with 48 GB of memory. We choose a batch size of 8, and set hyperparameters $L = 112$ across all experiments. We set the model bank size of $K = 5$ to balance performance and memory usage. For evaluation, we use the KITTI benchmark's official metrics, reporting average precision for car class in both 3D (*i.e.*, AP$_{3D}$) and bird's eye view (*i.e.*, AP$_{BEV}$), over 40 recall positions, with a 0.7 IoU threshold. We set S$_{feat}$ to a small positive value $\epsilon = 0.01$ once $\text{rank}(\cdot)$ reaches $D$, to ensure $\tilde{\mathbf{G}}$ is invertible. The detection model is pretrained using the training set of the source dataset, and subsequently adapted and tested on the validation set of KITTI.

**Augmentations.** We adopt data augmentation strategies from prior studies (Yang et al., 2022; 2021; Luo et al., 2021; Chen et al., 2023a) for methods requiring augmentations, such as MemCLR (VS et al., 2023), CoTTA (Wang et al., 2022a), and MOS. While CoTTA suggests employing multiple augmentations, our empirical results indicate that for TTA-3OD, utilizing only a single random world scaling enhances performance, whereas additional augmentations diminish it. Consequently, following the approach (Luo et al., 2021), we implement random world scaling for the mean-teacher baselines, applying strong augmentation (scaling between 0.9 and 1.1) and weak augmentation (scaling between 0.95 and 1.05) for all test-time domain adaptation tasks.

**Pseudo-labeling.** We directly apply the pseudo-labeling strategies from (Yang et al., 2021; 2022) to CoTTA and MOS for self-training, using the default configurations.

**Baseline Losses.** For Tent (Wang et al., 2021) and SAR (Niu et al., 2023), which calculate the entropy minimization loss, we sum the losses based on classification logits for all proposals from the first detection stage. For MemCLR (VS et al., 2023), we integrate its implementation into 3D detectors by reading/writing pooled region of interest (RoI) features extracted from the second detection stage, and compute the memory contrastive loss. For all baseline methods, we use default hyperparameters from their implementation code.

**Ensemble Details and Memory Optimization in MOS.** In the proposed MOS, we optimize memory usage by sequentially loading each checkpoint from the hard disk onto the GPU, extracting the output intermediate feature maps and box predictions needed for calculating the gram matrix and synergy weights. After saving the outputs for the currently loaded checkpoint, we remove it from the GPU before loading the next one. Once the synergy weights are calculated, in the model assembly process, we again sequentially load checkpoints, multiplying each model's parameters by the corresponding synergy weight and aggregate them into the supermodel. We report the optimized memory usage in Fig. 7.

Table 10: TTA-3OD results (easy/moderate/hard AP$_{3D}$) of **cyclist** class under the cross-corruption scenario (Waymo → KITTI-C) at heavy corruption level.

| | No Adaptation | Tent (Wang et al., 2021) | CoTTA (Wang et al., 2022a) | SAR (Niu et al., 2023) | MemCLR (VS et al., 2023) | **MOS** |
|---|---|---|---|---|---|---|
| Fog | 21.15/17.91/16.66 | 23.62/19.21/18.33 | 22.60/18.74/17.57 | 23.49/19.02/18.10 | 23.43/19.01/17.86 | **25.62/21.34/20.10** |
| Wet. | 60.36/49.61/47.20 | 59.72/48.57/45.96 | 61.36/49.27/47.04 | 57.43/46.48/43.96 | 57.76/46.34/44.79 | **64.73/52.11/49.60** |
| Snow | 48.87/40.37/37.96 | 52.81/42.25/40.11 | 52.91/41.55/39.09 | 52.09/41.89/39.26 | 52.24/41.71/39.28 | **57.12/45.53/43.39** |
| Moti. | 34.62/29.25/27.33 | 36.79/29.04/27.31 | 40.37/31.18/29.34 | 37.53/29.78/28.12 | 38.19/29.75/28.19 | **43.90/34.63/32.87** |
| Beam. | 32.48/22.42/21.26 | 36.08/25.03/23.85 | 30.89/21.37/20.42 | 37.16/26.21/24.74 | 36.34/25.35/24.32 | **43.06/30.19/28.43** |
| CrossT. | 59.56/48.75/46.20 | 58.72/49.26/46.51 | 62.14/49.13/46.21 | 58.66/49.16/46.61 | 58.68/48.87/46.40 | **65.10/51.67/49.30** |
| Inc. | 59.62/49.03/46.82 | 59.14/47.87/45.11 | 59.89/47.62/45.44 | 58.86/47.93/45.51 | 58.91/48.28/45.68 | **60.55/49.57/47.23** |
| CrossS. | 18.38/11.40/10.93 | 24.84/15.04/14.66 | 20.98/12.77/12.28 | 26.19/15.29/14.95 | 25.46/15.06/14.49 | **30.34/18.66/17.73** |
| Mean | 41.88/33.59/31.79 | 43.96/34.53/32.73 | 43.89/33.95/32.17 | 43.93/34.47/32.66 | 43.88/34.30/32.63 | **48.80/37.96/36.08** |

---

**Algorithm 1** The Proposed Model Synergy for TTA-3OD

---

**Input:** $f_0$: source pretrained model, $\{x_t\}_{t=1}^T$: point clouds to test, **F**: model bank.
**Output:** $f_t$: model adapted to the target point clouds.
    **Phase 1: Warm-up**.
    Start self-training with $f_0$, and add each of trained $f_t$ into **F** until reaching capacity.
    **Phase 2: Model Synergy**.
    **for** each remaining batch $x_t$ **do**
        Calculate the inverse of generalized Gram matrix $\tilde{\mathbf{G}}^{-1}$, via Eq. (5), Eq. (8), Eq. (9), Eq. (10)
        Calculate synergy weights $\tilde{\mathbf{w}}$ with $\tilde{\mathbf{G}}^{-1}$, for each of model in **F**, via Eq. (5)
        Assemble models within **F** weighted by $\tilde{\mathbf{w}}$, into a super model $f_t^*$, via Eq. (6)
        Generate prediction for $x_t$ as pseudo label $\hat{\mathbf{B}}^t$ by $f_t^*$, to train the current model $f_t$, via Eq. (7)
        **Phase 3: Model Bank Update**.
        **for** each time of training/inferring $L$ online batches **do**
            Calculate the mean synergy weights $\bar{\mathbf{w}}$, via Eq. (11)
            Remove the most redundant model in **F** and add $f_t$ into **F**, via Eq. (12)
        **end for**
    **end for**

---

### A.5 EXPERIMENTAL RESULTS ACROSS MULTIPLE CLASSES

**Cross-Dataset Shifts**. We conduct experiments to evaluate the proposed test-time adaptation method across various datasets for multiple object classes. As shown in Tab. 8, our proposed MOS consistently surpasses all test-time adaptation (TTA) baselines in AP$_{BEV}$ for every class. Specifically, in the Waymo → KITTI task, MOS exceeds the leading baseline (CoTTA) by 3.3% for pedestrians and 2.6% for cyclists in AP$_{3D}$. For the more challenging nuScenes → KITTI adaptation, which faces a significant environmental shift (beam numbers: 32 vs. 64), MOS significantly outperforms CoTTA by 4.9% for pedestrians and 31.5% for cyclists, highlighting CoTTA's limitations with more severe domain shifts. Notably, MOS also surpasses the unsupervised domain adaptive (UDA) 3D object detection method (ST3D) by 11.4% in the cyclist class for the task of Waymo → KITTI, demonstrating its effectiveness across all object classes and competitive performance against the UDA approach.

**Cross-Corruption Shifts.** Our experiments further explore adapting to 3D scenes under the challenging cross-corruption shifts for pedestrian and cyclist classes. As shown in Tab. 9, our MOS method surpasses all baselines across most corruptions, achieving the highest mean AP$_{3D}$: 40.41, 36.26 and 33.24 in easy, moderate and hard difficulty, respectively. However, for motion blur, beam missing, and cross sensor, the effectiveness of pseudo-labelling methods such as CoTTA and MOS is reduced. This is due to the small size and rareness of pedestrian objects, which lead to less accurate pseudo labels. For cyclists, which are slightly larger objects than pedestrians, CoTTA improves AP$_{3D}$ in motion blur scenarios (CoTTA's 31.18 vs. MemCLR's 29.75) but still struggles with harder corruptions including cross sensor and beam missing, as indicated in Tab. 10. Conversely, MOS effectively addresses both cross sensor and beam missing challenges for cyclists, demonstrating superior results: 14.9% and 18.6% higher than the best-performing baseline, respectively. Despite pseudo labelling's limitations with rare and small-sized objects under severe corruptions, MOS presents a comparable test-time adaptation ability for pedestrian objects and shows the leading performance for cyclists, demonstrating the effectiveness of leveraging diverse category knowledge from historical checkpoints.

Table 11: Results of TTA-3OD on adapting Waymo → KITTI using **PV-RCNN** (Shi et al., 2020) and **DSVT** (Wang et al., 2023a). $AP_{BEV}/AP_{3D}$ are reported. Note that, MemCLR requires 2nd stage RoI features, thus inapplicable to the one-stage DSVT.

| Model | No Adapt | Tent | CoTTA | SAR | MemCLR | MOS |
|---|---|---|---|---|---|---|
| PV-RCNN | 63.60/22.01 | 55.96/27.49 | 67.85/38.52 | 59.77/21.33 | 55.92/15.77 | **72.60/52.45** |
| DSVT | 65.06/27.14 | 63.94/31.07 | 66.63/34.51 | 66.12/37.45 | –/– | **77.38/57.41** |

## A.6 VISUALIZATION OF SYNERGY WEIGHTS

In Fig. 9, we visualize the predicted point clouds from different checkpoints in a bank **F** of size $K = 3$, each annotated with its synergy weight (SW) by Eq. 5. We note that CKPT 2 detects more pedestrians (row 2, column 2) that are overlooked by others. To capture this knowledge, the proposed MOS assigns CKPT 2 with a higher SW when inferring the right point cloud (column 2). However, for inferring the left point cloud (column 1), CKPT 2 fails to localize the car detected by CKPT 3, resulting in CKPT 3 receiving a higher SW than CKPT 2. Regarding CKPT 1, since other checkpoints cover all its box predictions, it is assigned the lowest SW to minimize redundancy. Therefore, for each input point cloud, the proposed MOS employs a separate strategy (*i.e.*, weighted averaging by SW) to combine checkpoints, ensuring the acquisition of non-redundant, long-term knowledge from historical checkpoints.

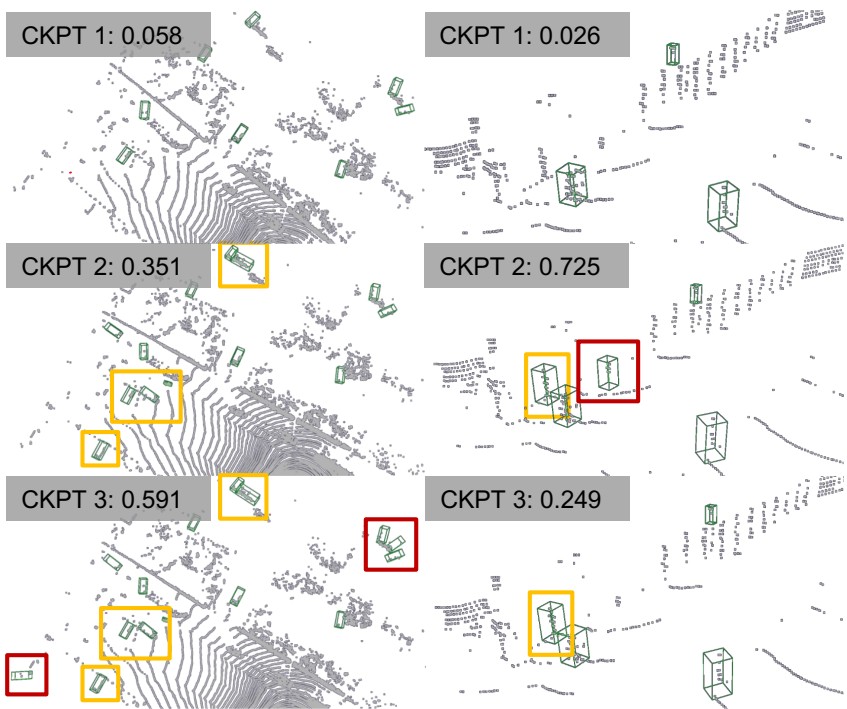

Figure 8: Visualization of box predictions from different checkpoints in the model bank of size $K = 3$, with corresponding synergy weight. These checkpoints are selected and stored in the model bank with the proposed MOS when adapting the 3D detector (Yan et al., 2018) pretrained on Waymo (Sun et al., 2020), to the KITTI (Geiger et al., 2012) dataset at test-time. Objects uniquely detected by only one checkpoint are marked in **red**, while those detected by two checkpoints are marked in **yellow**.

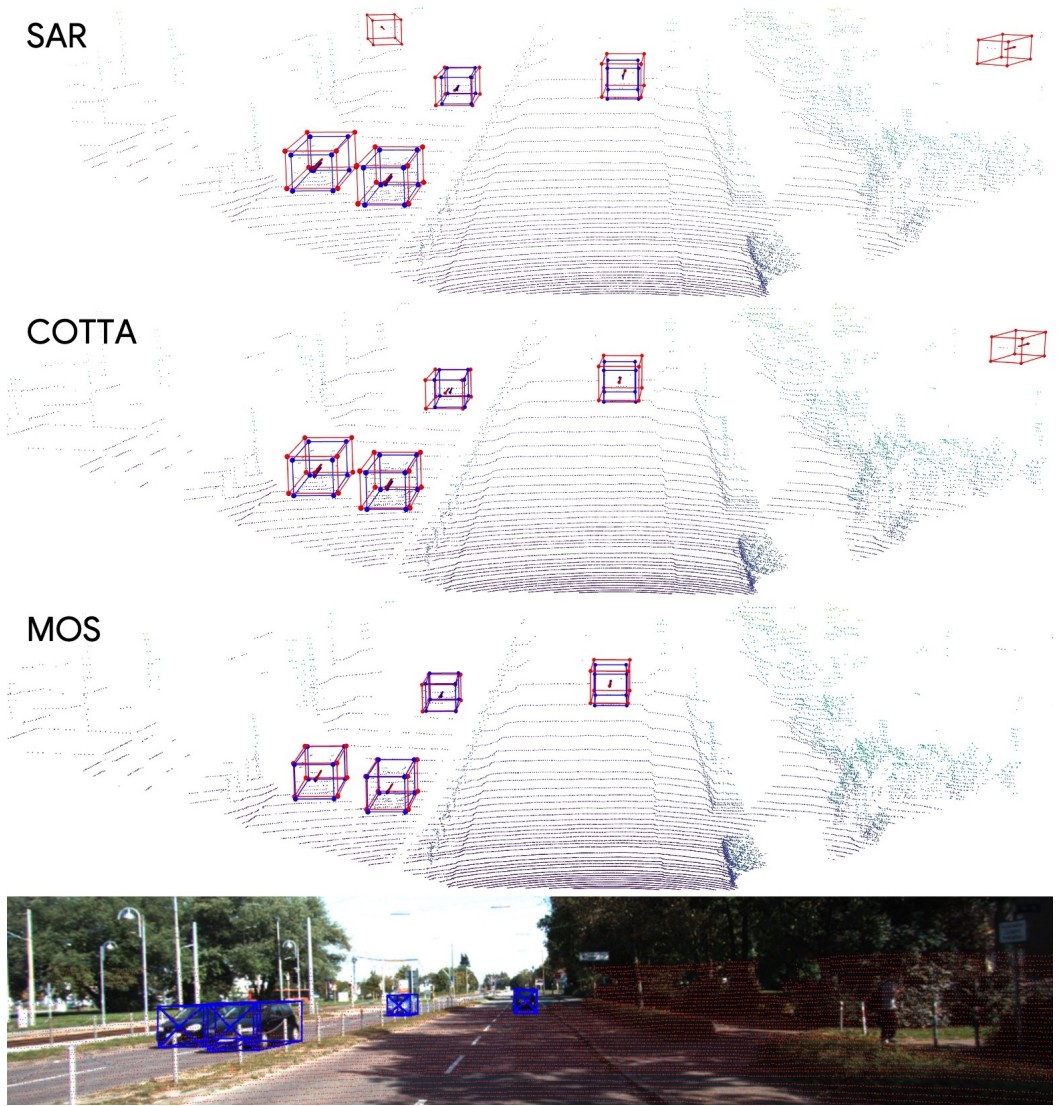

Figure 9: Visualization of box predictions from baseline methods and the proposed MOS, when adapting from Waymo (Sun et al., 2020) to KITTI (Geiger et al., 2012) at test time. Predicted boxes are shown in **red**, while ground truth boxes are shown in **blue**.

### A.7 SENSITIVITY TO 3D BACKBONE DETECTOR

We evaluate the sensitivity of the proposed MOS when integrated with a different two-stage, point-and voxel-based backbone detector: PVRCNN (Shi et al., 2020), and a **recently proposed**, one-stage, voxel transformer-based 3D detector DSVT (Wang et al., 2023a). We report the results of TTA baselines and the MOS for test-time adapting the PVRCNN and DSVT from Waymo to KITTI in Tab. 11. It is observed that MOS consistently outperforms the leading baseline, achieving substantial improvements of 36.16% and 53.48% in $AP_{3D}$ for PVRCNN and DSVT, respectively. This highlights that our model synergy framework is **model-agnostic**, maintaining its effectiveness across different 3D detectors.

### A.8 QUALITATIVE STUDY

In this section, we present visualizations of the box predictions generated by three different test-time adaptation (TTA) methods: 1) the optimization-based method SAR, 2) the mean-teacher-based

method CoTTA, and 3) the proposed MOS. The image at the botton is included for reference only. It is evident that both SAR and CoTTA exhibit false negatives, misclassifying background regions as objects. Additionally, even the correctly predicted boxes from SAR and CoTTA are poorly aligned with the ground truth. In contrast, the proposed MOS method produces no false negatives, and its predicted boxes show a high degree of overlap with the ground truth, resulting in a higher Intersection over Union (IoU) score.

