# OpenReview forum: "MOS: Model Synergy for Test-Time Adaptation on LiDAR-Based 3D Object Detection"
_ICLR.cc/2025/Conference — ICLR 2025 Oral_

### Official Review · Reviewer_REaw · 2024-10-29

**Soundness:** 3
**Presentation:** 3
**Contribution:** 3
**Rating:** 8
**Confidence:** 4

**Summary:**

This work proposes MOS, a framework that dynamically weights a sequence of models to create a more robust model based on the richness of new knowledge. MOS is evaluated in a test-time adaptation setting for LiDAR-based 3D object detection tasks. Experiments conducted in cross-dataset, cross-environment and cross-dataset&environment settings demonstrate the effectiveness of MOS.

**Strengths:**

- The concept of weighting checkpoints based on newly acquired knowledge is intriguing.
- The cross-dataset &environment setting can positively impact this field.
- This work is well-supported by experiments and ablation studies.

**Weaknesses:**

- Additional visualization of detection results is needed, particularly a comparison between methods that do not use new-knowledge-based model aggregation and MOS.
- From Table 9, the reviewer observes that the pedestrian detection results are unsatisfactory, and from Table 8, the reviewer notes that the cyclist detection results could be improved. This could benefit from adding 2D supervision, as rich contextual and color information in 2D images can facilitate the detection of small objects. A similar approach is suggested in recent work: Approaching Outside: Scaling Unsupervised 3D Object Detection from 2D Scene (ECCV '24). The authors are encouraged to include relevant references and discussions.

**Questions:**

- From Figure 7, the reviewer notes that both the time and memory costs of MOS are increased. Do the authors have any ideas for optimizing this additional cost?
- The reviewer wonders about the average time cost of conducting one experiment.

---

> ### Author Response · Authors · 2024-11-21
>
> We greatly appreciate your thoughtful review and valuable questions. Below, we have provided detailed responses to address each of your comments and concerns.
>
> > **Weakness 1**: Additional visualization of detection results is needed, particularly a comparison between methods that do not use new-knowledge-based model aggregation and MOS.
>
> Thanks for the questions! We have added a qualitative study with visualized 3D box predictions from different methods **in the revised manuscript, Appendix A.8**. You can find the visualized results **in Figure 9, page 23 of the uploaded manuscript**.
>
> We present visualizations of the box predictions generated by three different test-time adaptation (TTA) methods: 1) the optimization-based method SAR, 2) the mean-teacher-based method CoTTA, and 3) the proposed MOS. The image at the bottom is included for reference only. It is evident that both SAR and CoTTA exhibit false negatives, misclassifying background regions as objects. Additionally, even the correctly predicted boxes from SAR and CoTTA are poorly aligned with the ground truth. In contrast, the proposed MOS method produces **no false negatives**, and its predicted boxes show a **high degree of overlap with the ground truth**, resulting in a higher Intersection over Union (IoU) score.
>
> > **Weakness 2**: From Table 9, the reviewer observes that the pedestrian detection results are unsatisfactory, and from Table 8, the reviewer notes that the cyclist detection results could be improved. This could benefit from adding 2D supervision, as rich contextual and color information in 2D images can facilitate the detection of small objects. A similar approach is suggested in recent work: Approaching Outside: Scaling Unsupervised 3D Object Detection from 2D Scene (ECCV '24). The authors are encouraged to include relevant references and discussions.
>
> Thank you for highlighting this relevant and important work. After reading the paper, we found it to be truly impressive and of high quality. This work addresses the challenge of unsupervised 3D object detection without any ground truth labels. Existing methods often struggle with detecting distant or small objects due to the intrinsic limitations of LiDAR data, such as low spatial resolution and inherent sparsity. To address these challenges, the authors propose LiDAR-2D Self-paced Learning (LiSe), which integrates LiDAR and 2D image data. By leveraging the rich texture and localization information from RGB images, their approach improves the detection of distant and small objects. The method uses self-paced learning to refine pseudo-labels and employs adaptive sampling to balance learning across diverse object sizes and distances.
>
> The task setting of this paper differs from our test-time adaptation (TTA) for 3D object detection, as we focus on pure LiDAR-based 3D detectors, without integrating additional information from RGB images. However, as the reviewer correctly pointed out, our approach has not achieved comparable performance for small objects, such as pedestrians and cyclists, as it has for larger objects like cars. This is primarily due to the inherent sparsity of LiDAR data for small objects, which often results in very few or even just a handful of points, making it challenging for a purely LiDAR-based detector to learn effectively.
>
> In our future work, we will extend our task setting to multi-modal 3D object detectors, several modules from LiSe could be instrumental in improving the detection of small objects. For instance, the pretrained open-vocabulary 2D detector and segmentation model (e.g., SAM) can effectively refine the contours of small objects and optimize pseudo-labels by projecting them into 3D space. Additionally, the Adaptive Sampling Strategy used in LiSe could help balance the learning process across objects with varying sparsity and distances, further enhancing TTA performance.
>
> We again appreciate your attention to this outstanding work. We would like to note that the ECCV 2024 papers were released late this year, approaching the ICLR submission deadline, which is why we initially missed it. In our revised paper, we will make sure to cite this work and incorporate all the discussion from this rebuttal. Thank you once again!

---

> ### Author Response · Authors · 2024-11-21
>
> > **Question 1**: From Figure 7, the reviewer notes that both the time and memory costs of MOS are increased. Do the authors have any ideas for optimizing this additional cost?
>
>
> Thank you for your insightful question. In response to **Reviewer gqhp's Question 2**, we have clarified that the main bottleneck of our method does not stem from the algorithm itself. However, there are still potential strategies we can consider from a methodological perspective.
>
> To further reduce computational costs, some widely used strategies in previous TTA methods could potentially benefit 3D object detection tasks. One such strategy is **early stopping** [A, B], where for a new target domain, a signal can be used to halt model parameter updates. Subsequent test samples are then directly inferred thus saving overall adaptation time. In our proposed MOS framework, such a signal could be derived from the distribution of synergy weights in the model bank's checkpoints. If, for recent test batches, the distribution of synergy weights tends to be uniform, it indicates that the K checkpoints in the bank are sufficiently diverse, and further updates to the model bank are unnecessary. Consequently, the assembled super model can be directly used without any additional training process.
>
> Another commonly used strategy is **selectively choosing test batches for training** [C, D]. In our setup, it is essential to identify which test batches are worth learning from. For this purpose, we can estimate the average box prediction confidence for each test point cloud. If the prediction confidence is sufficiently high, it indicates that the model is capable of making highly accurate predictions. Further training could lead to overconfidence, then we can skip training on the current batch.
>
> We again appreciate the reviewer’s question, as it has inspired us to consider extending MOS with potential strategies to further accelerate test-time adaptation in our future work.
>
> [A] Press, O., Schneider, S., Kümmerer, M., & Bethge, M. (2024). Rdumb: A simple approach that questions our progress in continual test-time adaptation. Advances in Neural Information Processing Systems, 36.
>
> [B] Li, J., Jing, M., Su, H., Lu, K., Zhu, L., & Shen, H. T. (2021). Faster domain adaptation networks. IEEE Transactions on Knowledge and Data Engineering, 34(12), 5770-5783.
>
> [C] Niu, S., Wu, J., Zhang, Y., Wen, Z., Chen, Y., Zhao, P., & Tan, M. (2023). Towards stable test-time adaptation in dynamic wild world. arXiv preprint arXiv:2302.12400.
>
> [D] Yoo, J., Lee, D., Chung, I., Kim, D., & Kwak, N. (2024). What How and When Should Object Detectors Update in Continually Changing Test Domains?. In Proceedings of the IEEE/CVF Conference on Computer Vision and Pattern Recognition (pp. 23354-23363).
>
> > **Question 2** The reviewer wonders about the average time cost of conducting one experiment.
>
>
> Thanks! The following table presents the time cost for the proposed MOS in conducting experiments across all transfer tasks evaluated in this work:
>
> | Transfer Tasks  | Time Cost (minutes) |
> |---------|----------|
> | Waymo → KITTI    |  23:06   |
> | nuScenes → KITTI     |  24:26    |
> | KITTI → KITTI-C  |  22:28   |
> | Waymo → KITTI-C   |  22:59   |
> | Waymo →  nuScenes  | 41:42     |
>
> All experiments were conducted using a single NVIDIA RTX A6000 with the SECOND backbone 3D detector. The slightly higher time cost for the nuScenes → KITTI transfer is due to the nuScenes model configuration having a larger processing range([-75.2, -75.2, -2, 75.2, 75.2, 4] compared to KITTI [0, -40, -3, 70.4, 40, 1]). We will add this table to the revised manuscript. Thanks again!

---

> > ### Comment · Reviewer_REaw · 2024-11-25
> >
> > The author’s rebuttal addresses the reviewer’s concerns, and the reviewer is satisfied with the improvements made to the paper. As a result, the reviewer has raised the paper's score accordingly.

---

> > > ### Author Response · Authors · 2024-11-25
> > > **Thank You for Reading and Consideration**
> > >
> > > Dear Reviewer REaw,
> > >
> > > Thank you for your positive response and for raising your score! We are glad that our clarifications addressed your concerns and will ensure they are included in the final version.
> > >
> > > Best,
> > >
> > > Authors.

---

### Official Review · Reviewer_gqhp · 2024-11-02

**Soundness:** 3
**Presentation:** 3
**Contribution:** 2
**Rating:** 8
**Confidence:** 3

**Summary:**

The paper explores the test-time adaptation for LiDAR-based 3D object detection and proposes a model synergy approach that leverages long-term knowledge from historical checkpoints to tackle this task. Experiments are shown to verify the approach by tested against existing test-time adaptation strategies across three datasets.

**Strengths:**

-The paper focuses on exploring test-time adaptation for 3D object detection and proposes a natural and effective model synergy strategy that dynamically assigns weights to historical checkpoints and aggregates knowledge to tackle task.
-The paper analyzes and discusses the effectiveness of the synergy weights and makes task-specific modifications.
-Experiments verifies the method's effectiveness under various conditions, outperforming comparison methods.
-The paper is well-writing and easy to follow.

**Weaknesses:**

The theory on which the method is based has been considered in prior works. The paper makes minor modifications specific to the 3D object detection task.
A few minor clarifications are required in approach introduction， I have elaborated in the following section

**Questions:**

To maintain feature diversity, the method assigns higher weights to checkpoints with unique insights. This may lead to more false positive results; how does the method address this issue?

Does improving accuracy at the cost of time and computational resources go against the original intent of test-time adaptation?

---

> ### Author Response · Authors · 2024-11-21
>
> We greatly appreciate your valuable comments and thoughtful questions. Below, we have provided detailed responses to address each of your concerns.
>
> > **Weakness**: The theory on which the method is based has been considered in prior works. The paper makes minor modifications specific to the 3D object detection task. A few minor clarifications are required in approach introduction, I have elaborated in the following section.
>
> Thank you for your question! Our proposed TTA method is specifically designed for 3D object detectors. In **Sections 3.1.3** and **3.1.4**, we introduce two metrics tailored for comparing the similarity of outputs from a pair of 3D object detectors:
>
> 1. **Global BEV Feature Map Similarity:** This metric compares the overall features of the input 3D point cloud, including both object and background environmental features.
> 2. **Box Prediction Similarity with Hungarian Matching:** This metric evaluates the similarity of 3D detection boxes by considering the regression values for box coordinates, size, and orientation, as well as the classification of object categories.
>
> These two similarity metrics, **specifically designed for 3D features**, enable the MOS to select highly diverse historical checkpoints of 3D detection models.
>
> > **Question 1**: To maintain feature diversity, the method assigns higher weights to checkpoints with unique insights. This may lead to more false positive results; how does the method address this issue?
>
> Thanks! Our method follows the pipeline of previous unsupervised domain adaptation (UDA) approaches (IoU-based Quality-aware Criterion and Triplet Box Partition module) [A, B, C, D] for 3D detection to generate **high-quality** pseudo-labeled 3D boxes for self-training. Therefore, the quality of both the 3D boxes compared (Section 3.1.4) and the final produced 3D boxes (Equation 7) is ensured. Leveraging our assembled super model, which carries long-term and diverse knowledge, the final predictions achieve both **reliability** and **diversity**.
>
> [A] Yang, J., Shi, S., Wang, Z., Li, H., & Qi, X. (2021). St3d: Self-training for unsupervised domain adaptation on 3d object detection. In Proceedings of the IEEE/CVF conference on computer vision and pattern recognition (pp. 10368-10378).
>
> [B] Yang, J., Shi, S., Wang, Z., Li, H., & Qi, X. (2022). St3d++: Denoised self-training for unsupervised domain adaptation on 3d object detection. IEEE transactions on pattern analysis and machine intelligence, 45(5), 6354-6371.
>
> [C] Chen, Z., Luo, Y., Wang, Z., Baktashmotlagh, M., & Huang, Z. (2023). Revisiting domain-adaptive 3D object detection by reliable, diverse and class-balanced pseudo-labeling. In Proceedings of the IEEE/CVF International Conference on Computer Vision (pp. 3714-3726).
>
> [D] Hu, Q., Liu, D., & Hu, W. (2023). Density-insensitive unsupervised domain adaption on 3d object detection. In Proceedings of the IEEE/CVF Conference on Computer Vision and Pattern Recognition (pp. 17556-17566).
>
> > **Question 2**: Does improving accuracy at the cost of time and computational resources go against the original intent of test-time adaptation?
>
> Thanks for this good question! We fully agree with the reviewer that test-time adaptation (TTA) methods should prioritize efficiency. In fact, the **core design** of our approach is **highly efficient**. The feature-level similarity computation (Section 3.1.3) is performed using `torch.linalg.matrix_rank` on the GPU, enabling high-speed processing. The box similarity computation (Section 3.1.3) leverages Hungarian matching, which typically has a complexity of $O(n^3)$ [E]. However, since the number of boxes generated for each point cloud is usually in the range of a few to a few dozen, the Hungarian matching computation is very fast. Regarding the cost in computing the inverse of the Gram matrix (Equation 4), which we accomplish with `torch.linalg.inv` on the GPU. Although this has a typical complexity of $O(n^3)$, our $K$ values are very small (e.g., 3, 5, 7), ensuring quick computation.
>
> The main bottleneck in computation is not inherent in our method but rather in loading each checkpoint model to the GPU for every incoming test batch (details in Appendix A.4). Specifically, for each frame, running Equations (4), (5), (8), and (10) takes only **0.04** seconds, whereas loading the three checkpoints takes **0.18** seconds.
>
> Thus, the time spent loading checkpoints is **not caused by the computing methodology** we have proposed. In real-world deployment, this becomes an engineering challenge. From a methodological perspective, we have proposed two potential acceleration strategies in response to **Reviewer REaw Question 1**.
>
>
> [E] Mills-Tettey, G. A., Stentz, A., & Dias, M. B. (2007). The dynamic hungarian algorithm for the assignment problem with changing costs. Robotics Institute, Pittsburgh, PA, Tech. Rep. CMU-RI-TR-07-27.

---

> > ### Comment · Reviewer_gqhp · 2024-11-26
> >
> > Thank you for your response and answer my concerns.  I have raised my score accordingly. I have no further questions for the authors.

---

> > > ### Author Response · Authors · 2024-11-26
> > > **Thank You for Reading and Consideration**
> > >
> > > Dear Reviewer gqhp,
> > >
> > > Thank you for your positive response and for raising your score! We are glad that our clarifications addressed your concerns.
> > >
> > > Best,
> > >
> > > Authors

---

### Official Review · Reviewer_rufS · 2024-11-04

**Soundness:** 3
**Presentation:** 4
**Contribution:** 4
**Rating:** 8
**Confidence:** 3

**Summary:**

This paper introduces a novel approach for test-time adaptation in LiDAR-based 3D object detection: Model Synergy (MOS). The proposed method dynamically selects and assembles historical checkpoints to create a composite "super model" that adapts to domain shifts, including cross-dataset and corruption shifts. The assembly of checkpoints is controlled by Synergy Weights (SW) module, which minimizes redundancy by leveraging output-level and feature-level similarities. The method addresses practical scenarios like cross-corruption shifts, where both dataset and environmental discrepancies occur simultaneously. Experimental results across multiple datasets, including KITTI, Waymo, and KITTI-C, demonstrate that MOS outperforms SOTA test-time adaptation methods, achieving considerable improvements in both standard and complex scenarios.

**Strengths:**

The paper introduces a novel weighted ensemble of historical checkpoints approach, that helps mitigate catastrophic forgetting while adapting models to unseen domain shifts. This method achieves SOTA performance across multiple datasets and scenarios.
The proposed method also effectively reduces memory usage by dynamically updating the model bank, retaining only key models with high synergy weights, thus maintaining performance with low resource cost

**Weaknesses:**

The framework proposed by this paper is generally novel and effective, however, I still have some questions about the approach:
1. The paper demonstrates strong results (outperforms the SOTA methods) for LiDAR-based 3D object detection but does not extend its findings to other modalities or sensors. I'm interested in whether MOS can generalize to other types of data, such as inputs combining LiDAR point clouds and camera images.
2. While the paper contains empirical results for the chosen similarity metrics, more theoretical analysis for determining the synergy weights could be more thoroughly discussed.
3. It would be better if the authors could provide more clarity on how checkpoints are selected and what "unique insights" would improve understanding of the model selection process, I may not quite follow what's the insights here.
4. The rank-based feature-level and output-level similarity functions calculate the inverse Gram matrix, this method is novel but seems to be computational heavily. The paper provides some analysis of the MOS method in the last section, but it'll be great to have more discussion on how these similarity metrics scale with larger datasets or different types of models.

**Questions:**

Most questions are mentioned in the weakness section above.
Whether MOS can be generalized to other types of data, such as inputs combining LiDAR point clouds and camera images?
How checkpoints are selected and what "unique insights" would improve understanding of the model selection process?
How does the computational cost of calculating the synergy weights scale with the size of the model bank and batch size?
Also, I'm curious about if there's any evaluation of the impact of the proposed method on adaptation scenarios involving different types of LiDAR sensors or point cloud resolutions.
I appreciate the response to these questions.

---

> ### Author Response · Authors · 2024-11-21
>
> Thank you for your constructive review comments and insightful questions! We have provided responses below to address each of your concerns.
>
> > **Weakness 1**: The paper demonstrates strong results (outperforms the SOTA methods) for LiDAR-based 3D object detection but does not extend its findings to other modalities or sensors. I'm interested in whether MOS can generalize to other types of data, such as inputs combining LiDAR point clouds and camera images.
>
>
> Thanks for the question! The proposed MOS can be seamlessly integrated into most multi-modal 3D detection models, as it measures similarity between 3D models based on output feature maps and predicted 3D bounding boxes. As detailed in Sections 3.1.3 and 3.1.4 of the paper, MOS can measure the similarities between a pair of multi-modal 3D detectors and select historical checkpoints to assemble a "super model" that offers effective supervision.
>
> However, there is a significant **difference** between multimodal detectors and LiDAR-only detectors, as the former involve additional complexities, such as fusion and interaction between different modalities [A, B], and the extraction of temporal features from consecutive frames [C]. The proposed MOS is **specifically designed for LiDAR-based 3D detectors**, and therefore does not account for the unique properties of multimodal inputs. Inspired by your comment, we recognize this as a promising research direction and plan to extend MOS in future work to adapt multimodal models at test time.
>
>
> [A] Liu, Z., Tang, H., Amini, A., Yang, X., Mao, H., Rus, D. L., & Han, S. (2023, May). Bevfusion: Multi-task multi-sensor fusion with unified bird's-eye view representation. In 2023 IEEE international conference on robotics and automation (ICRA) (pp. 2774-2781). IEEE.
>
> [B] Yin, J., Shen, J., Chen, R., Li, W., Yang, R., Frossard, P., & Wang, W. (2024). Is-fusion: Instance-scene collaborative fusion for multimodal 3d object detection. In Proceedings of the IEEE/CVF Conference on Computer Vision and Pattern Recognition (pp. 14905-14915).
>
> [C] Li, X., Fan, B., Tian, J., & Fan, H. (2024). GAFusion: Adaptive Fusing LiDAR and Camera with Multiple Guidance for 3D Object Detection. In Proceedings of the IEEE/CVF Conference on Computer Vision and Pattern Recognition (pp. 21209-21218).
>
> > **Weakness 2**: While the paper contains empirical results for the chosen similarity metrics, more theoretical analysis for determining the synergy weights could be more thoroughly discussed.
>
> Thanks! One of the most relevant studies [D] highlights that when merging parameters, those that are influential for one model but redundant for others may cause influential values to be obscured by redundant ones, ultimately lowering overall model performance. This insight aligns with our motivation to assign lower synergy weights to checkpoints that carry duplicate knowledge. However, their method only considers the similarity between parameters, whereas our approach is focused on the model's output perspective. Given that parameter similarity among 3D models does not directly correlate with detection outcomes, we designed two similarity metrics specifically targeting the output of 3D detectors to ensure that diverse information is captured from the same point cloud.
>
> Furthermore, a comprehensive review of deep learning model ensembles [E] discusses the Regression Mean (RegMean) method, which aims to minimize the ℓ2 distance between the merged model and multiple models trained on different datasets.  This also **aligns with our assumption** in Section 3.1.1 that the ensemble super model remains unbiased towards any single model. However, the difference is that RegMean is multiple model trained on different datasets while **our method suits the TTA setting** where each model is trained on different test batches.
>
> In summary, these related model merging studies provide sufficient analysis consistent with the motivation and assumptions of our proposed MOS. The similarity metrics chosen in our approach are more suitable for 3D object detection tasks.
>
> [D] Yadav, P., Tam, D., Choshen, L., Raffel, C. A., & Bansal, M. (2024). Ties-merging: Resolving interference when merging models. Advances in Neural Information Processing Systems, 36.
>
> [E] Li, W., Peng, Y., Zhang, M., Ding, L., Hu, H., & Shen, L. (2023). Deep model fusion: A survey. arXiv preprint arXiv:2309.15698.

---

> ### Author Response · Authors · 2024-11-21
>
> > **Weakness 3**: It would be better if the authors could provide more clarity on how checkpoints are selected and what "unique insights" would improve understanding of the model selection process, I may not quite follow what's the insights here.
>
> **How checkpoints are selected?**: The K checkpoints are **dynamically updated within the model** bank during the test time adaptation. As illustrated in Figure 4 (Phase 1), during inference/adaptation of the **first K batches**, we store each checkpoint sequentially to construct a model bank. for **every L batches** inferred/adapted, we record the synergy weight of each checkpoint for each batch, then compute the average synergy weight over these L batches (Equation 11, Phase 2 in Figure 4). The checkpoint with the minimum average synergy weight (i.e., high similarity to other checkpoints) is then removed to reduce redundancy in the model bank, and the new checkpoint is added (Equation 11, Phase 3 in Figure 4). By doing so, the model bank is consistently updated to retain only K checkpoints while minimizing redundancy. For each incoming batch, all checkpoints in the model bank are assembled into a super model based on synergy weights.
>
>
> **What "unique insights?"**: Consider an autonomous vehicle driving along a road, where many small-sized cars are present. The 3D detection model trained on these recent LiDAR frames may become biased towards smaller-sized vehicles. If a larger vehicle such as a bus or an uncommon object like a bicycle suddenly appears in the scene, the model on the most recent checkpoint will struggle to detect it effectively. In such situations, our MOS aims to **retrieve insights from earlier checkpoints that have learned unique features** about buses or bicycles, thereby improving the prediction performance.
>
> > **Weakness 4**: The rank-based feature-level and output-level similarity functions calculate the inverse Gram matrix, this method is novel but seems to be computational heavily. The paper provides some analysis of the MOS method in the last section, but it'll be great to have more discussion on how these similarity metrics scale with larger datasets or different types of models.
>
>
> Thank you for the question! Our proposed MOS is **agnostic to dataset size**, and the **adaptation speed** (i.e., seconds per frame) **only depends on the model bank size K**. For each upcoming test batch $i$, we utilize $K$ historical models from the model bank (with $K$ empirically set to 3) to perform inference on the batch, resulting in $K$ inferences. We then compute a $K \times K$ similarity matrix, where each element is a combination of feature-level similarity (Section 3.1.3) and bounding box-level similarity (Section 3.1.4). Consequently, the adaptation and inference speed for a single point cloud is influenced only by $K$. Refer to Figure 6: using $K=7$ yields only a marginal performance improvement of 3.2% compared to $K=3$, while the **speed significantly decreases from 0.49 seconds/frame to 1.25 seconds/frame**. This highlights a trade-off between performance and computational cost. We opted for the smallest value of $K=3$ that balances these factors effectively.
>
> Regarding **different model types**, TTA methods are inherently influenced by model complexity. For instance, PV-RCNN is **significantly heavier** than SECOND, resulting in slower adaptation speeds for **all** TTA methods. When switching from SECOND to PV-RCNN, the adaptation time per frame for TENT [A] increases from 0.26 to 0.45 seconds, and for MOS, it increases from 0.49 to 0.88 seconds. A potential solution is to update only a small subset of parameters, or select parts of test samples to train [B, C], which can significantly improve training speed. Further discussions on this are included in the **rebuttal response to Reviewer REaw's Question 1**. Thank you again for your question! We will incorporate these discussions into the revised paper.
>
>
> [A] Wang, D., Shelhamer, E., Liu, S., Olshausen, B., & Darrell, T. (2020). Tent: Fully test-time adaptation by entropy minimization. arXiv preprint arXiv:2006.10726.
>
> [B] Niu, S., Wu, J., Zhang, Y., Wen, Z., Chen, Y., Zhao, P., & Tan, M. (2023). Towards stable test-time adaptation in dynamic wild world. arXiv preprint arXiv:2302.12400.
>
> [C] Yoo, J., Lee, D., Chung, I., Kim, D., & Kwak, N. (2024). What How and When Should Object Detectors Update in Continually Changing Test Domains?. In Proceedings of the IEEE/CVF Conference on Computer Vision and Pattern Recognition (pp. 23354-23363).

---

> ### Author Response · Authors · 2024-11-21
>
> > **Question**: I'm curious about if there's any evaluation of the impact of the proposed method on adaptation scenarios involving different types of LiDAR sensors or point cloud resolutions.
>
> Thank you for your question! In our experiments, we have incorporated **different types of LiDAR sensors** and **various point cloud resolutions**. For instance, in **Table 1**, the experiment adapting from nuScenes to KITTI demonstrates a test-time adaptation from sparser point clouds (extracted by 32-beam sensor) to denser point clouds (extracted by 64-beam sensor), indicating adapting form **low resolution to  high resolution**. The results show that our proposed method, MOS, outperforms the best baseline by 7.4%, highlighting its ability to effectively address the domain gap caused by different types of LiDAR sensors and resolustions.
>
> Furthermore, in **Appendix 2 (Table 7)**, the experiment adapting from Waymo to nuScenes represents the **opposite case**, transitioning from the denser 64-beam Waymo point clouds to the lower-resolution nuScenes data. Even in this scenario, our method consistently outperforms all baselines. These results demonstrate the robustness and effectiveness of our method for both **low-to-high** and **high-to-low resolution adaptations**.

---

> > ### Comment · Reviewer_rufS · 2024-11-26
> >
> > Thank you for addressing my concern in detail. I believe this paper is well presented, and I will maintain my original score. Thank you for your excellent work!

---

> > > ### Author Response · Authors · 2024-11-27
> > > **Thank You for Reading and Consideration**
> > >
> > > Dear Reviewer rufS,
> > >
> > > Thank you for your positive response! We extend our appreciation once again for your recognition of our work!
> > >
> > > Best,
> > >
> > > Authors.

---

### Author Response · Authors · 2024-11-21
**A Summary of Reviews**

Thank you to all reviewers for taking the time to review our paper and providing valuable and constructive comments! We are very grateful to all reviewers for giving our paper **positive ratings** and recognition in the following aspects:

1) The setting of this work is **positively influent** (reviewer REaw).
2) The proposed methodology is **novel**, **natural**, **effective** and **intriguing** (all reviewers).
3) The paper is **well-writing** and **easy to follow** (reviewer gqhp).
4) The experiments and ablation studies **well supported** and verified the proposed method across multiple datasets, conditions and scenarios, achieving **SOTA** performance (all reviewers).

Thanks once again to all reviewers for your valuable feedback on our paper! You can find our individual responses below your review comments. **If you have any further concerns or questions, we are more than willing to engage in a follow-up discussion with you!**

---

### Meta-Review · Area_Chair_P3FH · 2024-12-19

**Metareview:**

The paper proposes a test-time adaptation framework for LiDAR-based 3D object detection. The main idea is to dynamically select and assemble historical checkpoints to build a composite "super model" that adapts to domain shifts. Overall, the paper is well-written, and the idea is novel within the context of LiDAR-based 3D object detection. However, the efficiency of the proposed method should be improved to better align with the original intent of test-time adaptation, which emphasizes real-time performance and minimal computational overhead. Despite this limitation, all reviewers give positive scores. Based on the overall quality of the work, the novelty of the approach, and the feedback from the reviewers, the decision is to recommend the paper for acceptance, with the suggestion that the authors address the efficiency concerns in future work.

**Additional Comments On Reviewer Discussion:**

The paper was reviewed by three experts in the field and finally received all positive scores: 8, 8, and 8.
The major concerns of the reviewers are:
1.	some details of the method and extra experimental results,
2.	the generalizability to multimodal data,
3.	the efficiency.
The authors address most of the above concerns during the discussion period. Hence, I make the decision to accept the paper as poster.

---

### Decision · Program_Chairs · 2025-01-22

Accept (Oral)